# DuConTE: Dual-Granularity Text Encoder with Topology-Constrained Attention for Text-attributed Graphs

## Abstract

Text-attributed graphs integrate semantic information of node texts with topological structure, offering significant value in various applications such as document classification and information extraction. Existing approaches typically encode textual content using language models (LMs), followed by graph neural networks (GNNs) to process structural information. However, during the LM-based text encoding phase, most methods not only perform semantic interaction solely at the word-token granularity, but also neglect the structural dependencies among texts from different nodes. In this work, we propose DuConTE, a dual-granularity text encoder with topology-constrained attention. The model employs a cascaded architecture of two pretrained LMs, encoding semantics first at the word-token granularity and then at the node granularity. During the self-attention computation in each LM, we dynamically adjust the attention mask matrix based on node connectivity, guiding the model to learn semantic correlations informed by the graph structure. Furthermore, when composing node representations from word-token embeddings, we separately evaluate the importance of tokens under the center-node context and the neighborhood context, enabling the capture of more contextually relevant semantic information. Extensive experiments on multiple benchmark datasets demonstrate that DuConTE achieves state-of-the-art performance on the majority of them.

## 1 Introduction

Text-attributed graphs (Yang et al., 2021; Seo et al., 2024) have emerged as an increasingly significant research domain, with substantial applications in real-world scenarios such as social media analysis (Seo et al., 2024), academic citation systems (Wang et al., 2025), and knowledge base construction (Zhang et al., 2024). In such graphs, each node is associated with a piece of textual content, resulting in richly structured data that encapsulates both semantic text information and topological structure. Learning high-quality representations that effectively capture both the textual and structural characteristics of nodes is crucial for downstream tasks such as node classification (Zhao et al., 2024).

Recently, a growing body of research (Chen et al., 2023; Chien et al., 2021; Zhu et al., 2024) has begun leveraging Transformer-based language models (LMs) to model textual information in text-attributed graphs, aiming to enhance graph neural networks (GNNs). Thanks to their strong pretrained understanding of natural language, LMs can produce highly expressive representations of textual content. For example, GraphBridge(Wang et al., 2024) attempts to combine the text from the center-node and its neighbors into the LM, enabling the model to jointly encode the central text and its contextual information from neighboring nodes. Current approaches (Zhu et al., 2024; He et al., 2023; Jin et al., 2023) that jointly employ GNNs and LMs largely follow a common paradigm: the LM is responsible for encoding textual features, while the GNN focuses on capturing structural information.

However, existing approaches typically perform semantic interaction only at the word-token granularity when using LMs for text encoding, failing to capture meaningful node-granularity semantic interactions—where the textual content of different nodes is treated as holistic units and interacts

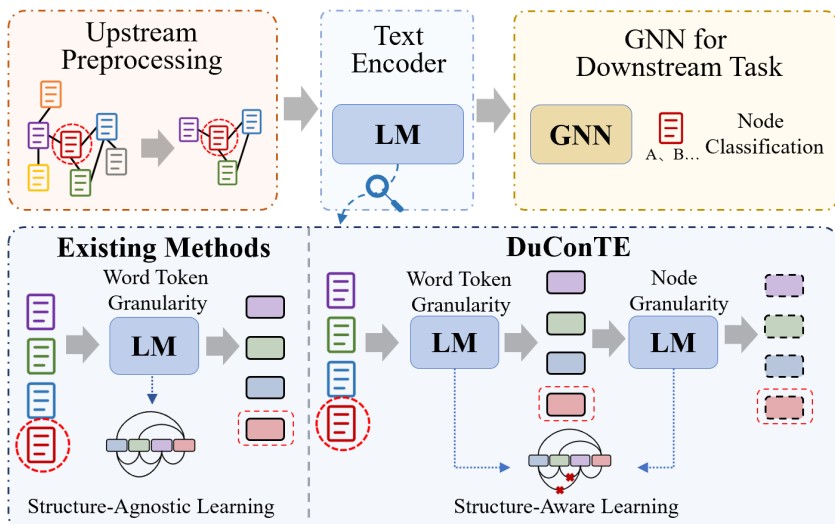

Figure 1: Overview of the text-attributed graph learning pipeline (top) and comparison between existing methods and the proposed DuConTE (bottom).

across the graph. Moreover, current methods either do not incorporate structural information into the LM at all, or the injected structural signals are insufficient to guide the encoding process effectively. Additionally, existing methods lack an effective mechanism for composing node representations from word-token embeddings.

To address these limitations, we propose **DuConTE**, a dual-granularity text encoder with topology-constrained attention for text-attributed graphs. As illustrated in the top panel of Figure 1, the text-attributed graph learning pipeline consists of three stages, with DuConTE acting as a plug-and-play text encoder module. It takes as input the text of each node and its sampled neighborhood structure (e.g., from random walks or k-hop sampling), obtained through upstream preprocessing, and outputs enriched node representations for downstream GNN models.

DuConTE performs **dual-granularity semantic encoding**, in which two pretrained LMs sequentially encode textual semantics at the **word-token** and **node** granularities, respectively. This design aligns with the inherent multi-granular nature of text-attributed graphs, allowing for a more complete capture of textual semantics. During the encoding process, DuConTE employs a **topology-constrained attention mechanism** to leverage graph structural information for enhanced text encoding. This is achieved through an attention masking strategy specifically designed for TAG, motivated by the homophily analysis in Section 6.3, enabling pretrained LMs to better process graph-structured textual data without architectural modification. Furthermore, we design a **node representation composer** that assesses the importance of individual word tokens under both **center-node** and **neighborhood** semantic contexts. This enables the model to capture salient semantic information more effectively when composing node representations from word-token embeddings.

- We propose **DuConTE**, a dual-granularity text encoder with topology-constrained attention for text-attributed graphs. It performs **dual-granularity semantic encoding** to model textual semantics at both the **word-token granularity** and **node granularity**, capturing a comprehensive, multi-scale understanding of the text-attributed graph.

- We introduce a **topology-constrained attention mechanism** that leverages an attention masking strategy, specifically designed for TAGs and grounded in the homophily analysis in Section 6.3, to effectively incorporate structural guidance into the textual encoding process.

- We design a **node representation composer** that distinctly models token importance under **center-node** and **neighborhood** contexts, enabling effective fusion of word-token embeddings into comprehensive node representations.

## 2 RELATED WORK

### 2.1 TEXT-ATTRIBUTED GRAPH LEARNING

Learning on text-attributed graphs has evolved from employing simple text features like Bag-of-Words (Zhang et al., 2010) to sophisticated methods centered on language models (LMs) (Chen et al., 2023; Chien et al., 2021; Zhu et al., 2024). These modern approaches generally follow two main paradigms. The first relies on a single, powerful LM to jointly process text and structure. For instance, LLaGA (Chen et al., 2024) injects structural information by mapping it into the LM's token space and relies solely on the LM to generate predictions. While conceptually unified, this paradigm is often computationally demanding, suffers from poor scalability, and achieves limited effectiveness in leveraging structural information. The second, more common paradigm, employs a hybrid LM-GNN pipeline where an LM first serves as a text encoder, and a subsequent GNN performs the downstream task using the resulting node embeddings. Representative works like GraphBridge (Wang et al., 2024) enrich node text with neighbor semantics before encoding, whereas Engine (Zhu et al., 2024) uses a GNN to process features from multiple LM layers. A critical limitation across most hybrid models is that the LM encoding process remains largely unaware of the graph topology. This decoupled approach hinders the deep fusion of structural and semantic information, a key challenge we address in this work.

### 2.2 TRANSFORMERS FOR MODELING STRUCTURED DATA

In recent years, numerous studies have leveraged Transformers to process graph-structured data (Shehzad et al., 2024). An early effort in this direction is Graph-BERT (Zhang et al., 2020), which applies a BERT-style Transformer to sampled subgraphs without relying on message passing. More recent approaches further enhance structural awareness: Graphormer (Ying et al., 2021) enhances the Transformer's understanding of graph structures by introducing spatial encoding and degree encoding. Another work NeuralWalker (Chen et al., 2025) generates serialized representations of graphs through random walks to exploit the self-attention mechanism of Transformers for modeling purposes. Edge-augmented methods (Rampášek et al., 2022; Satorras et al., 2021) explicitly model edge features to enhance the Transformer's sensitivity towards different edge types. Masked Graph Modeling (Hou et al., 2023; Tian et al., 2024) employs a masking strategy to learn structural information by predicting masked node or edge features. Notably, another strategy enhances structural awareness by using attention masks to explicitly control token interactions. K-BERT (Liu et al., 2020) employs a visibility mask to prevent injected knowledge tokens from attending to irrelevant input positions, preserving original semantics. UniD2T (Li et al., 2024) constructs attention masks based on the connectivity of a unified graph derived from structured data (e.g., tables, knowledge graphs) to enforce structure-aware interactions during pre-training. In this work, based on the homophily analysis in Section 6.3, we design a TAG-specific attention masking strategy to inject structural information at both word-token and node granularities.

## 3 PRELIMINARIES

### 3.1 PROBLEM FORMULATION

**Definition 1. Text-Attributed Graph.** A text-attributed graph (TAG) is formally defined as a triplet $\mathcal{G} = (\mathcal{V}, \mathcal{E}, \mathcal{T})$. Here, $\mathcal{V} = \{v_1, v_2, \ldots, v_N\}$ is the set of $N$ nodes, and $\mathcal{E} \subseteq \mathcal{V} \times \mathcal{V}$ is the set of edges describing the graph's topological structure, which can be represented by an adjacency matrix $\mathbf{A} \in \{0, 1\}^{N \times N}$. Each node $v_i \in \mathcal{V}$ is associated with a text description $\mathbf{w}_i$, and $\mathcal{T} = \{\mathbf{w}_1, \mathbf{w}_2, \ldots, \mathbf{w}_N\}$ denotes the collection of all node-associated text descriptions, where each $\mathbf{w}_i = (w_{i1}, w_{i2}, \ldots, w_{iL_i})$ is a sequence of word tokens of length $L_i$.

**Definition 2. Node Classification in Text-Attributed Graphs.** Given a text-attributed graph $\mathcal{G}$ and a set of $K$ predefined classes $\mathcal{C} = \{c_1, c_2, \ldots, c_K\}$, the task of node classification aims to learn a mapping function $f : \mathcal{V} \to \mathcal{C}$. The objective of this function is to predict the correct label $y_i \in \mathcal{C}$ for every node $v_i \in \mathcal{V}$ by jointly considering the graph structure $\mathcal{E}$ and the semantic information $\mathcal{T}$.

## 3.2 TRANSFORMER AND SELF-ATTENTION WITH MASKING

The Transformer architecture utilizes self-attention to capture dependencies within sequences. Given input $\boldsymbol{X} \in \mathbb{R}^{n \times d}$, query, key, and value projections are computed as $\boldsymbol{Q} = \boldsymbol{X W}_Q$, $\boldsymbol{K} = \boldsymbol{X W}_K, \boldsymbol{V} = \boldsymbol{X W}_V$. The process is:

$$\text{Attention}(\boldsymbol{Q}, \boldsymbol{K}, \boldsymbol{V}) = \text{softmax}\left(\frac{\boldsymbol{Q K}^\top}{\sqrt{d_k}} + \boldsymbol{M}\right) \boldsymbol{V}, \tag{1}$$

where $\boldsymbol{M}$ is derived from a binary mask matrix $\boldsymbol{M}_{mask} \in \{0, 1\}^{n \times n}$: valid attention positions are marked as $1$ in $\boldsymbol{M}_{mask}$, and their corresponding entries in $\boldsymbol{M}$ are set to $0$; invalid positions are marked as $0$ in $\boldsymbol{M}_{mask}$, and their entries in $\boldsymbol{M}$ are set to $-\infty$. This mechanism enables the model to selectively attend to semantic interactions between specific tokens, a property that we leverage to design our topology-constrained attention mechanism.

## 4 METHOD

In this section, we propose **DuConTE** illustrated in Figure 2, a dual-granularity text encoder with topology-constrained attention. It employs two language models as a word-token encoder $\mathcal{M}_L$ and a node encoder $\mathcal{M}_N$ respectively, both incorporating topology-constrained attention mechanisms. Given a target node $v_i$ and its neighborhood $\mathcal{N}(v_i)$, DuConTE first concatenates the textual content of $v_i$ and all nodes in $\mathcal{N}(v_i)$, and applies $\mathcal{M}_L$ to this combined sequence to generate word-token representations. A node representation composer then aggregates these into first-stage node representations. Subsequently, $\mathcal{M}_N$ encodes the sequence of first-stage node representations to produce a second-stage node representation for $v_i$. The final representation $\boldsymbol{o}_i$ is obtained through a weighted fusion of the node's first-stage and second-stage representations.

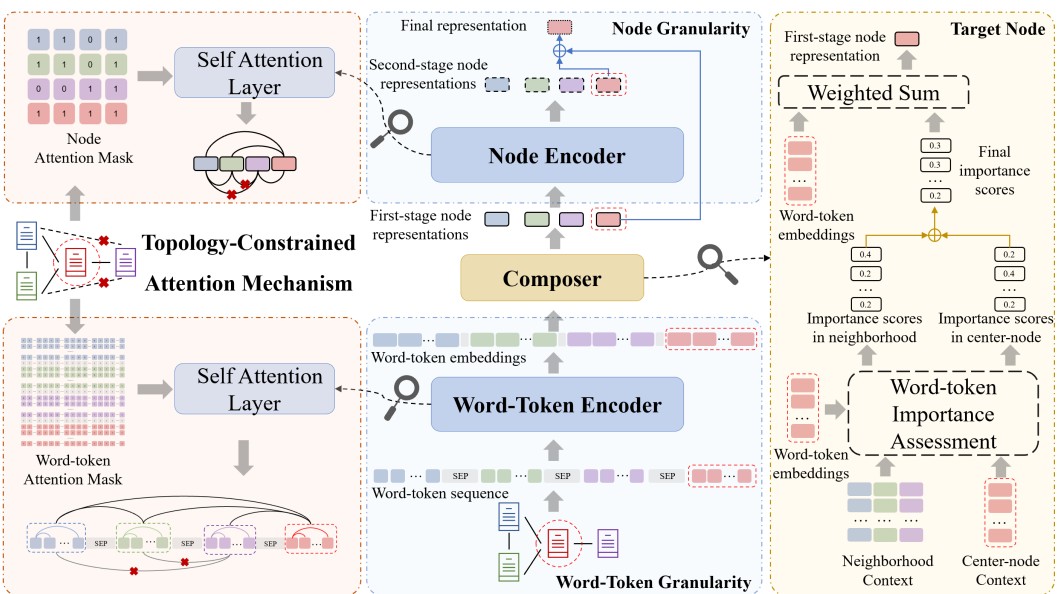

Figure 2: Overview of DuConTE with the dual-granularity cascaded architecture (middle), the topology-constrained attention mechanism (left), and the target node representation construction process in the node representation composer (right). The node representation composer is denoted as **Composer** in the figure.

## 4.1 DUAL-GRANULARITY SEMANTIC ENCODING

To capture semantics at the word-token and node granularities, which naturally exist in text graphs, we propose a dual-granularity cascaded architecture, illustrated in the middle of Figure 2. This architecture employs the word-token encoder $\mathcal{M}_L$ for the word-token granularity and the node encoder $\mathcal{M}_N$ for the node granularity, in a sequential manner.

**Word-Token Granularity Encoding.** Given a target node $v_i \in \mathcal{V}$ and its neighborhood $\mathcal{N}(v_i) \subseteq \mathcal{V}$, let $S^{(i)} = \{v_i\} \cup \mathcal{N}(v_i)$ denote the set consisting of the target node and its neighbors. For each node $v_j \in S^{(i)}$, we obtain its associated word-token sequence $\mathbf{w}_j = (w_{j1}, \ldots, w_{jL_j}) \in \mathcal{T}$. These sequences are concatenated with [SEP] tokens inserted between adjacent nodes to form a unified neighborhood input:

$$\mathbf{W}^{(i)} = [\mathbf{w}_{j_1}; [\text{SEP}]; \cdots; \mathbf{w}_{j_{|\mathcal{N}(v_i)|}}; [\text{SEP}]; \mathbf{w}_i] \in \mathbb{R}^{L \times d_L}, \tag{2}$$

where $v_{j_1}, \ldots, v_{j_{|\mathcal{N}(v_i)|}} \in \mathcal{N}(v_i)$.

The word-token encoder $\mathcal{M}_L$ (a pre-trained LM) processes $\mathbf{W}^{(i)}$ to perform semantic interaction at the word-token granularity, producing word-token embeddings $\boldsymbol{H}^{(i)} \in \mathbb{R}^{L \times d_L}$:

$$\boldsymbol{H}^{(i)} = \mathcal{M}_L(\mathbf{W}^{(i)}) = [\boldsymbol{h}_{j_1}^{(i)}; \boldsymbol{h}_{\text{SEP}_1}^{(i)}; \ldots; \boldsymbol{h}_i^{(i)}], \tag{3}$$

where $\boldsymbol{h}_j^{(i)} \in \mathbb{R}^{L_j \times d_L}$ is the embedding matrix for the tokens of node $v_j$ after such interaction, $\boldsymbol{h}_{\text{SEP}_k}^{(i)}$ denotes the embedding of the $k$-th [SEP] token, and $d_L$ is the hidden dimension of $\mathcal{M}_L$.

To distill these fine-grained word-token features into node semantics, we employ a node representation composer $f$, detailed in Section 4.3. This function maps $\boldsymbol{H}^{(i)}$ to a sequence of first-stage node representations $\boldsymbol{Z}^{(i)}$:

$$\boldsymbol{Z}^{(i)} = f\left(\boldsymbol{H}^{(i)}\right), \tag{4}$$

$$\boldsymbol{Z}^{(i)} = [\boldsymbol{z}_{j_1}^{(i)}; \ldots; \boldsymbol{z}_{j_{|\mathcal{N}(v_i)|}}^{(i)}; \boldsymbol{z}_i^{(i)}], \tag{5}$$

where each $\boldsymbol{z}_j^{(i)} \in \mathbb{R}^{d_L}$ denotes the first-stage node representation of $v_j$.

**Node Granularity Encoding.** To further model semantic interactions at the node granularity, we feed $\boldsymbol{Z}^{(i)}$ into node encoder $\mathcal{M}_N$ (another pre-trained LM), to produce a sequence of second-stage node representations $\boldsymbol{E}^{(i)}$:

$$\boldsymbol{E}^{(i)} = \mathcal{M}_N(\boldsymbol{Z}^{(i)}) \in \mathbb{R}^{(k+1) \times d_L}, \tag{6}$$

$$\boldsymbol{E}^{(i)} = [\boldsymbol{e}_{j_1}^{(i)}; \ldots; \boldsymbol{e}_{j_{|\mathcal{N}(v_i)|}}^{(i)}; \boldsymbol{e}_i^{(i)}], \tag{7}$$

where each $\boldsymbol{e}_j^{(i)} \in \mathbb{R}^{d_L}$ denotes the second-stage node representation of $v_j$.

Note that for $v_j \in \mathcal{N}(v_i)$, $\boldsymbol{z}_j^{(i)}$ and $\boldsymbol{e}_j^{(i)}$ are computed within the context of target node $v_i$, and thus represents a context-dependent, neighbor-oriented encoding—distinct from the representation obtained when $v_j$ is treated as a target node.

**Dual-Granularity Representation Fusion.** To integrate complementary semantic information from both granularities, we compute the final representation of the target node $v_i$ through a weighted combination of its first-stage and second-stage node representations:

$$\boldsymbol{o}_i = \alpha \cdot \boldsymbol{z}_i^{(i)} + (1 - \alpha) \cdot \boldsymbol{e}_i^{(i)}, \tag{8}$$

where $\alpha \in [0, 1]$ is a fixed fusion coefficient.

## 4.2 TOPOLOGY-CONSTRAINED ATTENTION MECHANISM

To endow our dual-granularity encoders with topological awareness, we transform their standard self-attention mechanism into a topology-constrained variant, as illustrated on the left in Figure 2. This is achieved through an attention masking strategy specifically designed for TAG. Informed by the homophily analysis in Section 6.3, it constructs masks based on node connectivity, applied at every layer and attention head to restrict attention exclusively between structurally connected word-tokens or nodes. The approach seamlessly integrates graph information without altering the core Transformer architecture.

**Word-Token Mask Construction.** For the word-token encoder $\mathcal{M}_L$ processing sequence $\mathbf{W}^{(i)} \in \mathbb{R}^{L \times d_L}$, we allow attention only between pairs of word-tokens within the same node or in connected nodes. Additionally, attention between [SEP] tokens and any word-token is always allowed to preserve a basic awareness of inter-node boundaries at the word-token granularity.

Accordingly, the attention mask matrix $\boldsymbol{M}_{mask}^{word}$ is constructed as follows: for any two tokens at positions $p$ and $q$ in $\mathbf{W}^{(i)}$, if neither token is a [SEP] token, let $v(p)$ and $v(q)$ denote their associated nodes in the graph. The entry $\boldsymbol{M}_{p,q}^{word} \in \{0,1\}^{L \times L}$ is defined as:

$$\boldsymbol{M}_{p,q}^{word} = \begin{cases} 1 & \text{if the token at } p \text{ or } q \text{ is } [\text{SEP}], \\ 1 & \text{if } v(p) = v(q) \text{ or } (v(p), v(q)) \in \mathcal{E}, \\ 0 & \text{otherwise}. \end{cases} \tag{9}$$

**Node Mask Construction.** For the node encoder $\mathcal{M}_N$ processing the sequence $\boldsymbol{Z}^{(i)} \in \mathbb{R}^{(k+1) \times d_L}$, we allow attention only between node representations that correspond to the same node or to connected nodes in the graph.

Accordingly, the attention mask matrix $\boldsymbol{M}_{mask}^{node}$ is constructed as follows: for any two positions $m$ and $n$ in $\boldsymbol{Z}^{(i)}$, let $v(m)$ and $v(n)$ denote the corresponding nodes in the graph. The entry $\boldsymbol{M}_{m,n}^{node} \in \{0,1\}^{(k+1) \times (k+1)}$ is defined as:

$$\boldsymbol{M}_{m,n}^{node} = \begin{cases} 1 & \text{if } v(m) = v(n) \text{ or } (v(m), v(n)) \in \mathcal{E}, \\ 0 & \text{otherwise}. \end{cases} \tag{10}$$

### 4.3 NODE REPRESENTATION COMPOSER

To effectively fuse the word-token embeddings $\boldsymbol{H}^{(i)}$ into high-quality first-stage node representations, we design a Node Representation Composer $f$. The composer employs two distinct modules: a more sophisticated module $f_1$ to compute the representation of the target node $v_i$, and a lightweight module $f_2$ to independently encode each neighbor node $v_j \in \mathcal{N}(i)$. This asymmetric design enables the target node to capture rich contextual information while ensuring efficient and undisturbed representation learning for neighbors.

**Target Node Representation Construction.** To capture the most salient semantics of the target node $v_i$ under both center-node and neighborhood context—and to explicitly balance their relative influence—we design $f_1$ to assess word-token significance from dual perspectives, as shown on the right in Figure 2. Specifically, $f_1$ employs a specialized attention mechanism to compute the importance of each word-token in the target node's text $\mathbf{w}_i$.

With learnable projection matrices $\boldsymbol{W}_Q, \boldsymbol{W}_K \in \mathbb{R}^{d_L \times d_L}$, we compute the queries $\boldsymbol{Q}^{(i)}$ as the projected embeddings of all word-tokens in the neighborhood, and the keys $\boldsymbol{K}^{(i)}$ as the projected embeddings of the target node's word-tokens:

$$\boldsymbol{Q}^{(i)} = \boldsymbol{H}^{(i)} \boldsymbol{W}_Q \in \mathbb{R}^{L \times d_L}, \tag{11}$$

$$\boldsymbol{K}^{(i)} = \boldsymbol{h}_i^{(i)} \boldsymbol{W}_K \in \mathbb{R}^{L_i \times d_L}. \tag{12}$$

As defined in 3.1, $w_{jp}$ is the $p$-th word-token in node $v_j$. The attention weight $a_{j,p,q}^{(i)}$ from $w_{jp}$ to $w_{iq}$ is computed using the scaled dot-product attention mechanism, with softmax normalization over all queries attending to $w_{iq}$.

The total importance of $w_{iq}$ is decomposed into two components:

- **Importance under center-node context**: $\alpha_q^{\text{cen}} = \sum_{p=1}^{L_i} a_{i,p,q}^{(i)}$;

- **Importance under neighborhood context**: $\alpha_q^{\text{neigh}} = \sum_{v_j \in \mathcal{N}(i)} \sum_{p=1}^{L_j} a_{j,p,q}^{(i)}$.

Each component is independently normalized via softmax to obtain $\mu_q^{\text{cen}}$ and $\mu_q^{\text{neigh}}$, which are fused into the final importance score $\mu_q$ using a fixed coefficient $\beta \in [0,1]$:

$$\mu_q = \beta \cdot \mu_q^{\text{cen}} + (1-\beta) \cdot \mu_q^{\text{neigh}}. \tag{13}$$

The final representation $\boldsymbol{z}_i^{(i)}$ is a weighted sum over the target node's word-token embeddings:

$$\boldsymbol{z}_i^{(i)} = \sum_{q=1}^{L_i} \mu_q \boldsymbol{h}_{i,q}^{(i)}. \tag{14}$$

**Neighbor Node Representation Construction.** To enable efficient encoding while preserving each neighbor's intrinsic semantic content, we design a lightweight module $f_2$ that employs local attention pooling. Given a neighbor node $v_j \in \mathcal{N}(i)$, an importance score $s_{j,p}$ is computed for each word-token embedding $\boldsymbol{h}_{j,p}^{(i)}$ via a learnable projection vector $\boldsymbol{w}_a \in \mathbb{R}^{d_L}$. After softmax normalization to obtain weights $\pi_{j,p}$, the first-stage representation of $v_j$ is computed as a weighted sum:

$$\boldsymbol{z}_j^{(i)} = \sum_{p=1}^{L_j} \pi_{j,p} \boldsymbol{h}_{j,p}^{(i)}. \tag{15}$$

### 4.4 Two-stage training procedure

We train DuConTE using a two-stage procedure. We first train $\mathcal{M}_L$ and $f_1$ to learn high-quality first-stage node representations, then train $\mathcal{M}_N$ and $f_2$ based on these representations. The full training procedure is detailed in Appendix B.

## 5 Experiments

### 5.1 Datasets

In this paper, we evaluate DuConTE for node classification on five widely-used datasets: Cora (Sen et al., 2008), CiteSeer (Giles et al., 1998), WikiCS (Mernyei & Cangea, 2007), ArXiv-2023 (He et al., 2023), OGBN-Products (Hu et al., 2020) and Ele-Photo (Yan et al., 2023). For detailed descriptions of each dataset, please refer to Appendix F.

### 5.2 Baselines

To evaluate the effectiveness of our proposed model, we employ several baseline models for comparison. For a detailed description of all baseline models, please refer to Appendix C. These baselines can be categorized into three main types:

- **Graph-Specific Models:** Models specifically designed and trained from scratch for graph-structured data, *e.g.*, NodeFormer (Wu et al., 2022), GraphFormers (Yang et al., 2021).
- **Pure LMs:** Language models that perform inference solely based on node texts while completely ignoring the graph structure, *e.g.*, BERT (Devlin et al., 2019), RoBERTa (Liu et al., 2019).
- **Recent TAG Methods:** Leading approaches that have demonstrated strong performance on text-attributed graph benchmarks, *e.g.*, GraphBridge (Wang et al., 2024), ENGINE (Zhu et al., 2024).

### 5.3 Experimental Settings

**Evaluation Task and Metric.** In this study, we focus on node classification as the downstream task for text-attributed graphs, and adopt classification accuracy as the evaluation metric.

**Implementation Details.** We instantiate a text-attributed graph learning pipeline, as illustrated in the top panel of Figure 1. DuConTE serves as the text encoder in this pipeline, implemented with two RoBERTa-base models serving as the word-token encoder and node encoder respectively. In the downstream phase, a two-layer GraphSAGE with a hidden dimension of 64 is employed as the GNN component. All methods are evaluated under a unified experimental protocol to ensure a fair comparison. Detailed configurations for model hyperparameters, upstream preprocessing, implementation settings of baseline methods, and training procedures are provided in Appendix D.

### 5.4 Performance Comparison and Discussions

We compare the performance of various models on text-attributed graph node classification, with results reported in Table 1. DuConTE achieves state-of-the-art performance on most datasets, outperforming the second-best method by 2.7% on ArXiv-2023 and 1.6% on Cora. The results demonstrate

Table 1: **Experiment results**: Mean accuracy and standard deviation over 10 runs with different random seeds. **Bold** indicates the best performance, underlined denotes the second-best, and '–' signifies that the method is not applicable to the dataset. "DuConTE" refers to the pipeline instance using DuConTE as the text encoder, as described in Section 5.3.

| Methods | Cora | CiteSeer | WikiCS | ArXiv-2023 | OGBN-Products | Ele-Photo |
|---|---|---|---|---|---|---|
| GraphFormers | 80.29 ± 1.74 | 71.84 ± 1.23 | 71.37 ± 0.35 | 63.14 ± 0.59 | 68.09 ± 0.57 | 78.16 ± 0.17 |
| NodeFormer | 88.24 ± 0.34 | 74.96 ± 0.61 | 75.56 ± 0.51 | 67.68 ± 0.47 | 67.37 ± 0.83 | 77.47 ± 0.04 |
| GraphSAGE | 87.42 ± 1.31 | 72.26 ± 1.21 | 76.91 ± 0.77 | 68.56 ± 0.53 | 70.56 ± 0.27 | 79.87 ± 0.26 |
| BERT | 79.63 ± 1.81 | 71.27 ± 1.11 | 77.96 ± 0.57 | 76.84 ± 0.09 | 76.45 ± 0.16 | 68.73 ± 0.13 |
| Sentence-BERT | 78.94 ± 1.43 | 72.93 ± 1.84 | 77.84 ± 0.06 | 77.41 ± 0.55 | 74.98 ± 0.15 | 68.47 ± 0.24 |
| RoBERTa-base | 78.37 ± 1.29 | 71.76 ± 1.23 | 76.86 ± 0.52 | 77.24 ± 0.19 | 76.03 ± 0.12 | 69.31 ± 0.19 |
| RoBERTa-large | 79.81 ± 1.37 | 72.31 ± 1.74 | 77.64 ± 0.95 | 77.81 ± 0.43 | 76.24 ± 0.35 | 71.46 ± 0.13 |
| GLEM | 87.59 ± 0.17 | 77.42 ± 0.68 | 78.23 ± 0.56 | 79.23 ± 0.17 | 76.04 ± 0.34 | 83.56 ± 0.54 |
| TAPE | 87.48 ± 0.76 | – | – | 80.04 ± 0.31 | **79.23 ± 0.13** | – |
| SimTeG | 86.74 ± 1.71 | 78.51 ± 1.04 | 79.73 ± 0.84 | 79.45 ± 0.53 | 76.43 ± 0.49 | 83.71 ± 0.26 |
| ENGINE | 87.61 ± 1.34 | 76.84 ± 1.41 | 77.92 ± 0.89 | 78.57 ± 0.19 | 77.68 ± 1.31 | 82.46 ± 0.10 |
| GraphBridge | 93.60 ± 0.98 | 88.62 ± 0.76 | 80.47 ± 0.26 | 86.43 ± 0.29 | 77.92 ± 0.27 | 89.23 ± 0.15 |
| DuConTE | **95.24 ± 0.79** | **89.45 ± 1.22** | **81.09 ± 0.43** | **90.31 ± 0.35** | 78.80 ± 0.10 | **91.89 ± 0.18** |

DuConTE's ability to produce high-quality, semantically rich node representations that effectively support downstream GNN models.

# 6 ANALYSIS

## 6.1 ABLATION STUDY

We conduct ablation studies to evaluate the three key innovations in DuConTE. The variants are defined in Appendix H, including **NoDual**, **NoMask-T/D/Both**, and **MeanPool/Center-Only/Neigh-Only/UnifiedContext**. All variants are evaluated under the same experimental setup.

As shown in Table 2, DuConTE outperforms all variants, confirming the effectiveness of its three key designs: (1) DuConTE surpasses **NoDual** by +0.8% on Cora and OGBN-Products, verifying that dual-granularity encoding aligns with the inherent semantic granularity of text-structured graphs and thus better captures rich semantic information. (2) Performance drops in **NoMask-T/D/Both** confirm that topology-constrained attention effectively injects structural information at both word-token and node granularities; notably, **NoMask-D** consistently outperforms **NoMask-T**, suggesting that structural information is critical even at the finest semantic granularity. (3) The lower performance of **MeanPool** further validates that importance-based weighted fusion captures key semantic information more effectively than uniform averaging. Gains over **Center-Only**, **Neigh-Only**, and **UnifiedContext** demonstrate that both center-node and neighborhood contexts are important for assessing word-token importance, and explicitly differentiating their distinct influences leads to more accurate semantic weighting.

Table 2: Ablation results on Cora, CiteSeer, and OGBN-Products

| Methods | Cora | CiteSeer | OGBN-Products |
|---|---|---|---|
| NoDual | 94.46 ± 0.76 | 89.22 ± 1.34 | 77.98 ± 0.38 |
| NoMask-T | 94.23 ± 0.76 | 88.84 ± 1.28 | 78.19 ± 0.13 |
| NoMask-D | 94.59 ± 0.58 | 88.86 ± 1.27 | 78.52 ± 0.15 |
| NoMask-Both | 94.10 ± 0.85 | 89.04 ± 0.99 | 78.40 ± 0.17 |
| MeanPool | 94.43 ± 0.94 | 88.57 ± 0.95 | 78.27 ± 0.12 |
| Center-Only | 95.13 ± 0.80 | 88.46 ± 1.20 | 78.17 ± 0.18 |
| Neigh-Only | 95.09 ± 0.74 | 88.71 ± 1.40 | 78.36 ± 0.15 |
| UnifiedContext | 95.09 ± 0.86 | 88.98 ± 1.10 | 78.56 ± 0.23 |
| DuConTE | **95.24 ± 0.79** | **89.45 ± 1.22** | **78.80 ± 0.10** |

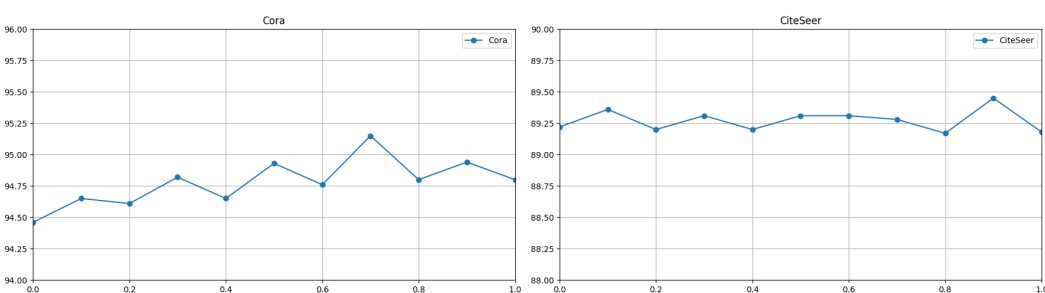

Figure 3: Sensitive analysis of the fusion coefficient $\alpha$

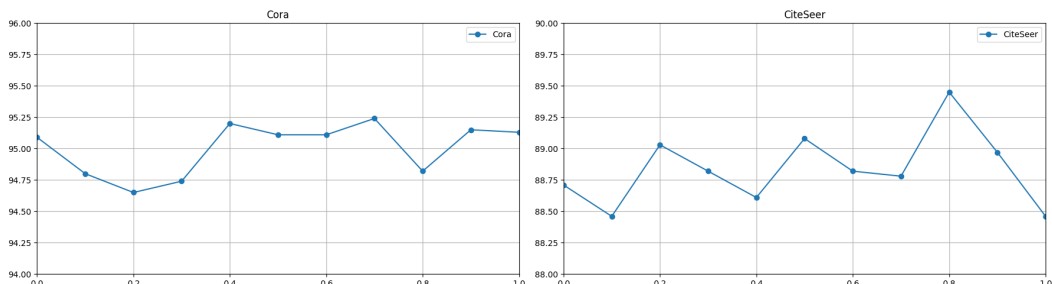

Figure 4: Sensitive analysis of the fusion coefficient $\beta$.

## 6.2 SENSITIVITY ANALYSIS

We analyze the sensitivity of DuConTE to the fusion coefficients $\alpha$ and $\beta$ over the range $[0, 1]$. The performance trends are shown in figure 3 and figure 4. Across all experiments, the performance variation remains within 1%, demonstrating the model's robustness to these hyperparameters.

For $\alpha$, which controls the fusion of dual-granularity semantic representations, the optimal performance on Cora and CiteSeer falls within the range $[0.7, 0.9]$. This indicates a clear fusion pattern: word-token granularity semantics provide stable and reliable information, while node granularity semantics contribute complementary yet essential signals—consistent with their role as more abstract, high-level features.

For $\beta$, which balances the influence of center-node and neighborhood contexts in word-token importance assessment, the performance trend varies across datasets, indicating that the relative importance of these two contexts is dataset-dependent. On Cora and CiteSeer, strong performance is observed within $[0.4, 0.7]$ and $[0.2, 0.8]$, respectively, confirming that both contexts contribute meaningfully. Notably, the optimal values consistently fall within $[0.6, 0.8]$, suggesting that the center-node context exerts a stronger influence—aligning with the intuition that a token's relevance is primarily shaped by the target node itself.

## 6.3 WHY TOPOLOGY-CONSTRAINED ATTENTION WORKS: A HOMOPHILY PERSPECTIVE

In this subsection, we analyze the effectiveness of topology-constrained attention from the perspective of the homophily assumption, which posits that connected nodes in a graph are more likely to share similar semantic properties. To the best of our knowledge, this assumption is well-supported by most widely used text-attributed graph benchmarks, where adjacent nodes are more likely to belong to the same class.This is further supported by the homophily statistics reported in Appendix G.

In the topology-constrained attention mechanism, the masks $M_{mask}^{token}$ and $M_{mask}^{node}$ are injected into the attention layers of the word-token encoder and the node encoder, respectively. As a result, cross-node attention interactions are constrained to occur between semantic information from connected nodes at both granularities. Under the homophily assumption, such information is more likely to be semantically related, thereby enabling mutually complementary interactions. This allows the model to effectively leverage the graph structure to learn higher-quality representations.

### 6.4 Additional Evaluation on Link Prediction

To assess the general applicability of DuConTE beyond node classification, we conduct link prediction experiments on the Cora, CiteSeer, and ArXiv-2023 datasets, using AUC as the evaluation metric. Detailed configurations and training procedures are provided in Appendix E. According to Table 3, DuConTE consistently outperforms baseline methods on the link prediction task, indicating that it is highly effective at representation learning on text-attributed graphs. This result further highlights the versatility of DuConTE and its potential for broader applications across diverse TAG-based tasks.

Table 3: Experimental Results on Link Prediction

| Methods | Cora | CiteSeer | ArXiv-2023 |
|---|---|---|---|
| GraphSAGE | $97.10 \pm 0.43$ | $87.29 \pm 1.22$ | $91.81 \pm 0.26$ |
| SimTeG | $97.86 \pm 0.44$ | $90.06 \pm 1.34$ | $93.12 \pm 0.46$ |
| GraphBridge | $98.07 \pm 0.77$ | $91.86 \pm 1.03$ | $94.35 \pm 0.65$ |
| DuConTE | $\mathbf{99.13 \pm 0.19}$ | $\mathbf{93.29 \pm 0.75}$ | $\mathbf{95.40 \pm 0.33}$ |

### 6.5 Parameter Efficiency Analysis

To evaluate the parameter efficiency of DuConTE, we replace the LM backbone in baseline methods with RoBERTa-large (340M parameters) while keeping other configurations unchanged. We then compare their performance against DuConTE using two RoBERTa-base models (150M parameters each) as its LM backbones. In this setup, every baseline has a larger total parameter count than DuConTE. TAPE is excluded from the comparison as it relies on a large language model. As shown in Table 4, DuConTE achieves the best performance despite using fewer parameters, highlighting its parameter efficiency. This suggests a novel parameter-efficient scaling paradigm: rather than improving performance by scaling up a single large LM, DuConTE achieves greater gains with fewer total parameters by leveraging two smaller LMs.

Table 4: **Experiment results**: Subscript $_{(large)}$ indicates the use of RoBERTa-large as the LM backbone, while $_{(base)}$ indicates RoBERTa-base.

| Methods | Cora | CiteSeer | WikiCS | ArXiv-2023 | OGBN-Products | Ele-Photo |
|---|---|---|---|---|---|---|
| GLEM$_{(large)}$ | $89.07 \pm 0.25$ | $78.04 \pm 0.36$ | $78.14 \pm 0.81$ | $78.94 \pm 0.45$ | $78.37 \pm 0.29$ | $84.73 \pm 0.67$ |
| SimTeG$_{(large)}$ | $88.64 \pm 0.89$ | $79.89 \pm 1.23$ | $80.16 \pm 0.65$ | $80.69 \pm 0.49$ | $78.31 \pm 0.61$ | $84.97 \pm 0.41$ |
| ENGINE$_{(large)}$ | $88.57 \pm 1.25$ | $78.14 \pm 0.74$ | $80.36 \pm 0.24$ | $77.37 \pm 0.43$ | $78.44 \pm 0.57$ | $83.43 \pm 0.23$ |
| GraphBridge$_{(large)}$ | $94.06 \pm 0.94$ | $88.91 \pm 0.98$ | $80.96 \pm 0.57$ | $87.14 \pm 0.36$ | $78.51 \pm 0.68$ | $90.96 \pm 0.19$ |
| DuConTE$_{(base)}$ | $\mathbf{95.24 \pm 0.79}$ | $\mathbf{89.45 \pm 1.22}$ | $\mathbf{81.09 \pm 0.43}$ | $\mathbf{90.31 \pm 0.35}$ | $\mathbf{78.80 \pm 0.10}$ | $\mathbf{91.89 \pm 0.18}$ |

### 6.6 Computational Overhead of the Node Representation Composer

We measure the training and inference time of DuConTE and its ablation variant **MeanPool** on Cora, CiteSeer, and Ele-Photo. As reported in Appendix I, the Node Representation Composer introduces an average overhead of 23.8% in training time and 19.9% in inference time. This cost is generally acceptable, and further acceleration is possible by reducing the dimensionality of keys and queries in $f_1$ to lower computational load. A key direction for future work is to design methods that convert word-token embeddings into node representations with both higher performance and lower computational cost. This is crucial for TAG representation learning but remains underexplored.

## 7 Conclusion

In this paper, we introduce **DuConTE**, a dual-granularity text encoder with topology-constrained attention for text-attributed graphs. DuConTE encodes node semantics at both word-token and node granularity to capture the inherent dual-granularity semantic structure of text-attributed graphs. Our topology-constrained attention mechanism utilizes an attention masking strategy specifically designed for TAG, offering an effective and architecture-preserving approach to adapt LMs to graph-structured data. In the node representation composer, the contexts of the center node and its neighborhood are separately considered to more effectively assess the semantic importance of word-tokens in the target node. Extensive experiments on multiple benchmark datasets show that DuConTE achieves state-of-the-art performance on the majority of them.

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

## AI Use Disclosure

The authors used ChatGPT (OpenAI, 2025) solely for English grammar and punctuation correction. No scientific content was generated or modified by the AI.

## A    Reproducibility Statement

**Dataset description.**    We provide a detailed description of the datasets, including information on their sources, in Appendix F. We describe the dataset splitting strategy in Appendix D.2.

**Baseline description.**    We provide a detailed description of the baseline models we used and include links to their source code in Appendix C.

**Implementation details.**    We provide a detailed description of the model hyperparameter settings and training configurations in Appendix D to facilitate reproducibility.

**Open access to code.**    The source code of DuConTE is included as a ZIP file in the supplementary materials. We will release it publicly via an open-source repository upon publication.

## B    Two-Stage Training Procedure of DuConTE

We train DuConTE using a two-stage procedure: the word-token encoder is trained first to learn high-quality representations, and the node encoder is then trained based on these representations.

**Stage 1: Word-Token Encoder Training.** We first train the word-token encoder $\mathcal{M}_L$ and the aggregator $f_1$, while the node encoder $\mathcal{M}_N$ and the aggregator $f_2$ are not involved in this stage. The first-stage representation of the target node, $\boldsymbol{z}_i^{(i)}$, serves as input to a learnable linear classifier $\mathbf{W}_{\text{cls}}^{(1)}$. The objective is to minimize the standard cross-entropy loss over the training set $\mathcal{V}_{\text{train}}$:

$$\mathcal{L}_1 = -\sum_{i \in \mathcal{V}_{\text{train}}} \boldsymbol{y}_i^\top \log(\text{softmax}(\mathbf{W}_{\text{cls}}^{(1)} \boldsymbol{z}_i^{(i)})). \tag{16}$$

**Stage 2: Node Encoder Training.** We then fix $\mathcal{M}_L$ and $f_1$, and train the node encoder $\mathcal{M}_N$ and the aggregator $f_2$. The final node representation $\boldsymbol{o}_i$ is fed to a new learnable classifier $\mathbf{W}_{\text{cls}}^{(2)}$ for prediction. The objective is to minimize the cross-entropy loss:

$$\mathcal{L}_2 = -\sum_{i \in \mathcal{V}_{\text{train}}} \boldsymbol{y}_i^\top \log(\text{softmax}(\mathbf{W}_{\text{cls}}^{(2)} \boldsymbol{o}_i)). \tag{17}$$

## C BASELINE

**Graph-Specific Models:** We adopt two graph transformers: GraphFormers (Yang et al., 2021)[Code] and NodeFormer (Wu et al., 2022)[Code]. We also adopt GraphSAGE (Hamilton et al., 2017)[Code], a Graph Neural Network, which also serves as the GNN backbone for other baseline models.

**Pure LMs:** We adopt four commonly used pre-trained language models: BERT (Devlin et al., 2019)[Code], Sentence-BERT (Reimers & Gurevych, 2019)[Code], and two versions of RoBERTa (Liu et al., 2019): RoBERTa-base[Code] and RoBERTa-large[Code].

**Recent TAG Methods:** **GLEM** (Zhao et al., 2022)[Code], is a framework that integrates language models and GNNs during training using a variational EM approach. **TAPE** (He et al., 2023)[Code], leverages large language models such as ChatGPT to generate pseudo labels and explanations for textual nodes. These are then used to fine-tune pre-trained language models alongside the original texts. **SimTeG** (Duan et al., 2023)[Code] uses a cascading structure specifically designed for textual graphs. It employs a two-stage training paradigm: first, it fine-tunes language models, and then it trains GNNs. **ENGINE** (Zhu et al., 2024)[Code] is an efficient fine-tuning and inference framework for text-attributed graphs. It co-trains large language models and GNNs using a ladder-side approach to optimize both memory and time efficiency. For inference, ENGINE utilizes an early exit strategy to further accelerate the process. **GraphBridge** (Wang et al., 2024)[Code] first encodes both local and global text information using a language model, by incorporating neighboring nodes' text. A GNN is then applied to further refine node representations.

## D NODE CLASSIFICTION: IMPLEMENTATION AND EXPERIMENTAL DETAILS

### D.1 COMPUTATIONAL RESOURCES

In our experiments, we use four NVIDIA GeForce RTX 3090 GPUs, each with 24 GB of VRAM. The LM components are trained and run on these four GPUs, while the GNN module is executed on a single GPU.

### D.2 DATASET SPLIT

For Cora and CiteSeer, we use a random node split with 60% of nodes for training, 20% for validation, and 20% for testing. For WikiCS, ArXiv-2023, and OGBN-Products, we adopt the official training, validation, and test splits (Mernyei & Cangea, 2007; He et al., 2023; Hu et al., 2020).

### D.3 BASELINE MODEL DEPLOYMENT SETTINGS

**Graph-Specific Models:** For NodeFormer and GraphSAGE, we use the raw node features from each dataset, constructed via one-hot encoding. For GraphFormers, we implement the model using its official source code.

**Pure LMs:** For BERT, Sentence-BERT, and RoBERTa-base, we perform full-parameter fine-tuning using the raw texts of each node. For RoBERTa-large, we employ Low-Rank Adaptation (LoRA) with a rank of 8.

**Recent TAG Methods:** We use RoBERTa-base as the language model backbone and a two-layer GraphSAGE with hidden size 64 as the GNN backbone. This configuration is consistent with that of DuConTE to ensure a fair comparison. We implement these models using their official source code, and the training epochs as well as learning rates for both the LM and GNN components are kept consistent with DuConTE.

### D.4 IMPLEMENTATION DETAILS OF OUR PIPELINE INSTANCE

We provide a comprehensive overview of the configuration and training parameters adopted by the pipeline instantiated in Section 5.3.

**Upstream Preprocessing Configurations.** We adopt 2-hop neighborhood sampling with a maximum of 39 neighbors per node. This means that for any node $v_i \in \mathcal{V}$, the sampled neighborhood $\mathcal{N}(v_i)$ satisfies $|\mathcal{N}(v_i)| \leq 39$, and we denote $S^{(i)} = \{v_i\} \cup \mathcal{N}(v_i)$ with $|S^{(i)}| \leq 40$.

The text of each node is processed using a reduction module (Wang et al., 2024) to fit the input length limit of the LM. This module, introduced in the GraphBridge framework, is a token selector pre-trained on the training set that assigns importance scores to word tokens within each node's text. Given that the RoBERTa-base model has a maximum context length of 512 tokens, we enforce a uniform token budget across all nodes in $S^{(i)}$. Specifically, let

$$B = \left\lfloor \frac{512}{|S^{(i)}|} \right\rfloor - 1$$

be the per-node token budget (excluding the `[SEP]` token). For any node $v_j \in S^{(i)}$ whose original token sequence $\mathbf{w}_j$ exceeds $B$ tokens, we retain only the top-$B$ most important tokens as ranked by the reduction module, preserving their original order. The resulting truncated sequences are then concatenated with `[SEP]` separators to form the unified input $\mathbf{W}^{(i)}$.

**Hyperparameter Settings of DuConTE.** For the internal hyperparameters $\alpha$ and $\beta$ of DuConTE, we perform a grid search over the range $[0, 1]$ with a step size of 0.1, selecting the best combination based on performance on the validation set. The selected hyperparameter values for each dataset are reported in Table 5.

Table 5: Hyperparameter settings of $\alpha$ and $\beta$ in the experiments.

| Hyperparameter | Cora | CiteSeer | WikiCS | ArXiv-2023 | OGBN-Products | Ele-Photo |
|:---:|:---:|:---:|:---:|:---:|:---:|:---:|
| $\alpha$ | 0.7 | 0.9 | 0.9 | 0.6 | 0.8 | 0.6 |
| $\beta$ | 0.7 | 0.8 | 0.8 | 0.9 | 0.9 | 0.7 |

**Training Setup for DuConTE.** DuConTE uses two pre-trained RoBERTa-base models for $\mathcal{M}_L$ and $\mathcal{M}_N$. $\mathcal{M}_L$ has positional encoding enabled. $\mathcal{M}_N$ takes $\boldsymbol{H}^{(i)}$ as input directly, bypassing the token embedding layer, with positional encoding kept on.

The detailed two-stage training procedure of DuConTE is described in Section B. In both Stage 1 and Stage 2, the learning rate is set to `5e−5`, and the number of training epochs is specified in Table 6.

**Training Setup for the Downstream GNN.** We adopt a two-layer GraphSAGE with a hidden dimension of 64 as the GNN backbone in the downstream task. The model is trained using the final node representations generated by DuConTE as input features. We employ a learning rate of `1e−2`, train for up to 500 epochs, and apply early stopping with a patience of 20 epochs based on validation performance.

Table 6: Training Epochs in Stage 1 and Stage 2

| Stage | Cora | CiteSeer | WikiCS | ArXiv-2023 | OGBN-Products | Ele-Photo |
|-------|------|----------|--------|------------|---------------|-----------|
| Stage 1 | 8 | 8 | 16 | 8 | 8 | 8 |
| Stage 2 | 8 | 8 | 16 | 8 | 8 | 8 |

# E  LINK PREDICTION: IMPLEMENTATION AND EXPERIMENTAL DETAILS

## E.1  DATASET SPLIT

For Cora, CiteSeer, and ArXiv-2023, we randomly split edges into training, validation, and test sets in a 6:2:2 ratio.

## E.2  BASELINE MODEL DEPLOYMENT SETTINGS

**GraphSAGE:**   We use a one-layer GraphSAGE with hidden dimension 16 and a two-layer MLP link predictor.

**Recent TAG Methods:**   We use RoBERTa-base as the language model backbone and a one-layer GraphSAGE with hidden dimension 16 as the GNN backbone, paired with a two-layer MLP link predictor. This configuration matches that of DuConTE to ensure a fair comparison. We implement these models using their official source code, and the training epochs as well as learning rates for both the LM and GNN components are kept consistent with DuConTE.

## E.3  IMPLEMENTATION DETAILS OF OUR PIPELINE INSTANCE

We instantiate a text-attributed graph learning pipeline for link prediction, with DuConTE serving as the text encoder. In the downstream phase, we use a one-layer GraphSAGE with hidden dimension 16 and a two-layer MLP link predictor.

**Upstream Preprocessing Configurations.**   We use the same upstream preprocessing configuration as in D.4.

**Hyperparameter Settings of DuConTE.**   The values of the internal hyperparameters $\alpha$ and $\beta$ are set as in Table 5.

**Training Setup for DuConTE.**   The training configuration of DuConTE follows that in D.4. The detailed training procedure is described in E.4.

**Training Setup for the Downstream GNN.**   We adopt a one-layer GraphSAGE with hidden dimension 16 as the downstream GNN, followed by a two-layer MLP link predictor, using the final node representations from DuConTE as input features. The model is trained with a learning rate of 1e−2, up to 500 epochs, and early stopping (patience = 20) based on validation performance.

## E.4  TWO-STAGE TRAINING PROCEDURE OF DUCONTE

We train DuConTE using a two-stage procedure tailored for link prediction. In both stages, link scores are computed as the dot product of node representations, and the model is optimized using binary cross-entropy loss on positive and negative edges.

**Stage 1: Word-Token Encoder Training.**   We train the word-token encoder $\mathcal{M}_L$ and the composer $f_1$, while $\mathcal{M}_N$ and $f_2$ remain frozen. For each training edge $(i, j) \in \mathcal{E}_{\text{train}}$, we compute the dot-product score between first-stage representations:

$$s_{ij}^{(1)} = (\boldsymbol{z}_i^{(i)})^\top \boldsymbol{z}_j^{(j)}.$$

A corresponding negative edge $(i, k)$ is sampled by replacing $j$ with a uniformly random node $k$. The loss is computed as:

$$\mathcal{L}_1 = \sum_{(i,j) \in \mathcal{E}_{\text{train}}} \left[ \ell(s_{ij}^{(1)}, 1) + \ell(s_{ik}^{(1)}, 0) \right], \tag{18}$$

where $\ell(\hat{y}, y) = \text{BCEWithLogits}(\hat{y}, y)$.

**Stage 2: Node Encoder Training.** We freeze $\mathcal{M}_L$ and $f_1$, and train $\mathcal{M}_N$ together with $f_2$. The final representations $\boldsymbol{o}_i$ and $\boldsymbol{o}_j$ are scored analogously:

$$s_{ij}^{(2)} = \boldsymbol{o}_i^\top \boldsymbol{o}_j.$$

Using the same positive/negative edge sampling strategy, the second-stage loss is:

$$\mathcal{L}_2 = \sum_{(i,j) \in \mathcal{E}_{\text{train}}} \left[ \ell(s_{ij}^{(2)}, 1) + \ell(s_{ik}^{(2)}, 0) \right]. \tag{19}$$

# F   DATASET DESCRIPTIONS

The experiments are conducted on five benchmark text-attributed graph datasets, widely adopted in graph representation learning. Below we provide a brief overview of each. For detailed statistics, including the number of nodes, edges, classes, and average token count per node, please refer to Table 7.

**Cora (Sen et al., 2008)**   The Cora dataset contains 2,708 scientific publications divided into seven classes: case-based reasoning, genetic algorithms, neural networks, probabilistic methods, reinforcement learning, rule learning, and theory. The papers form a citation network with 5,429 undirected edges, where each node has at least one citation link.

**CiteSeer (Giles et al., 1998)**   The CiteSeer dataset consists of 3,186 scientific documents categorized into six areas: Agents, Machine Learning, Information Retrieval, Databases, Human–Computer Interaction, and Artificial Intelligence. Each document is represented by its title and abstract, and the task is to classify papers based on this text and the citation structure.

**WikiCS (Mernyei & Cangea, 2007)**   WikiCS is a Wikipedia-based dataset for evaluating graph neural networks. It includes 10 classes corresponding to computer science topics and exhibits high connectivity. Node features are obtained from the text of the corresponding Wikipedia articles.

**ArXiv-2023 (He et al., 2023)**   ArXiv-2023 is a directed citation network introduced in TAPE, containing computer science papers from arXiv published in 2023 or later. Nodes represent papers, and directed edges represent citations. The task is to classify each paper into one of 40 subject areas, such as `cs.AI`, `cs.LG`, and `cs.OS`, using labels provided by authors and arXiv moderators.

**OGBN-Products (Hu et al., 2020)**   OGBN-Products is a dataset of Amazon products with co-purchase relations. The full version has over 2 million nodes and 61 million edges. The subset used here, created via node sampling in TAPE (He et al., 2023), contains 54,000 nodes and 74,000 edges. Each node corresponds to a product and is labeled with one of 47 top-level categories.

**Ele-Photo (Yan et al., 2023)**   Ele-Photo is a text-attributed graph derived from the AmazonElectronics dataset (Ni et al., 2019), where nodes represent electronics products and edges denote frequent co-purchases or co-views. Each node is assigned a label from a three-level hierarchy of electronics categories, with the task formulated as 12-way classification. The textual attribute of each node is constructed from the user review with the highest number of votes; if no such review exists, a random review is selected.

Table 7: **Dataset statistics**. **Nodes**, **Edges**, **Classes** and **Avg.degrees** mean the number of nodes, edges, classes and average degrees for each dataset, respectively. **Avg.tokens** represents the average number of tokens per node in each dataset when using the RoBERTa-base's tokenizer.

| Dataset | Nodes | Edges | classes | Avg.degrees | Avg.tokens |
|---|---|---|---|---|---|
| Cora | 2708 | 5492 | 7 | 3.90 | 194 |
| CiteSeer | 3186 | 4277 | 6 | 1.34 | 196 |
| WikiCS | 11701 | 215863 | 10 | 36.70 | 545 |
| ArXiv-2023 | 46198 | 78543 | 40 | 3.90 | 194 |
| OGBN-Products(subset) | 54025 | 74420 | 47 | 2.68 | 163 |
| Ele-Photo | 48362 | 500928 | 12 | 18.07 | 185 |

## G  HOMOPHILY ANALYSIS

In this section, we analyze the homophily of the six datasets used in our experiments: Cora (Sen et al., 2008), CiteSeer (Giles et al., 1998), WikiCS (Mernyei & Cangea, 2007), ArXiv-2023 (He et al., 2023), OGBN-Products (subset) (Hu et al., 2020) and Ele-Photo (Yan et al., 2023). Specifically, we compute the **label homophily ratio** $H$, defined as:

$$H = \frac{1}{|\mathcal{E}|} \sum_{(i,j)\in\mathcal{E}} \mathbb{I}(y_i = y_j), \tag{20}$$

where $\mathcal{E}$ denotes the set of edges, $y_i$ is the class label of node $i$, and $\mathbb{I}(\cdot)$ is the indicator function that equals 1 if the condition is true and 0 otherwise. This metric measures the proportion of edges connecting nodes with identical labels; a higher value indicates stronger homophily. The results are summarized in Table 8.

Table 8: Label Homophily Ratios Across Datasets

| Dataset | Cora | CiteSeer | WikiCS | ArXiv-2023 | OGBN-Products (subset) | Ele-Photo |
|---|---|---|---|---|---|---|
| Homophily ($H$) | 0.8100 | 0.7451 | 0.6547 | 0.6465 | 0.7950 | 0.7351 |

According to the results, all datasets exhibit homophily ratios above 0.6, indicating a relatively high level of homophily.

## H  ABLATION VARIANTS

In this section, we detail the design of each ablation variant used in our experiments.

**NoDual**  It encodes semantic information only at the word-token granularity, achieved by setting the hyperparameter $\alpha = 0$.

**NoMask-T**  It uses the vanilla self-attention mechanism in every attention layer of the word-token encoder.

**NoMask-D**  It uses the vanilla self-attention mechanism in every attention layer of the node encoder.

**NoMask-Both**  It uses the vanilla self-attention mechanism in every attention layer of both encoders.

**MeanPool**  It directly converts word-token embeddings into node representations using mean pooling.

**Center-Only**  Its node representation composer evaluates word-token importance only in the center-node semantic context, with the hyperparameter $\beta$ set to 1.

**Neigh-Only**  Its node representation composer evaluates word-token importance only in the neighborhood semantic context, with the hyperparameter $\beta$ set to 0.

**UnifiedContext**   Its node representation composer evaluates word-token importance in a shared context, without differentiating the contextual influence from the center-node and its neighborhood. The unnormalized importance of token $w_{iq}$ is computed as:

$$\mu'_q = \sum_{p=1}^{L_i} a_{i,p,q}^{(i)} + \sum_{v_j \in \mathcal{N}(i)} \sum_{p=1}^{L_j} a_{j,p,q}^{(i)}, \tag{21}$$

and the final importance score $\mu_q$ is obtained by applying softmax normalization over all word-tokens in $v_i$.

# I   COMPUTATIONAL OVERHEAD STATISTICS

We report the total training time (over 8 epochs) and single-pass inference time on the full dataset for DuConTE and its ablation variant MeanPool across Cora, CiteSeer, and Ele-Photo. All timing measurements were conducted on a system equipped with four NVIDIA GeForce RTX 4090 GPUs, each with 24GB of memory.

Table 9: Total Training Time (seconds)

| Method | Cora$_{(training)}$ | CiteSeer$_{(training)}$ | Ele-Photo$_{(training)}$ |
|---|---|---|---|
| DuConTE | 1054 | 434 | 5074 |
| MeanPool | 880 | 326 | 4278 |

Table 10: Total Inference Time (seconds)

| Method | Cora$_{(inference)}$ | CiteSeer$_{(inference)}$ | Ele-Photo$_{(inference)}$ |
|---|---|---|---|
| DuConTE | 185 | 62 | 796 |
| MeanPool | 163 | 49 | 663 |

