# OpenReview forum: "DuConTE: Dual-Granularity Text Encoder with Topology-Constrained Attention for Text-attributed Graphs"
_ICLR.cc/2026/Conference — Submitted to ICLR 2026_

### Official Review · Reviewer_pNQe · 2025-10-20

**Soundness:** 2
**Presentation:** 2
**Contribution:** 2
**Rating:** 2
**Confidence:** 5

**Summary:**

The paper introduces DuConTE, a novel model designed for text-attributed graphs, which integrate both semantic information from node texts and topological graph structure. Text-attributed graphs have significant applications in tasks such as document classification and information extraction. Traditional methods often use language models (LMs) to encode text and graph neural networks (GNNs) for processing the graph’s structural information. However, these methods usually focus on word-token granularity during the text encoding phase and fail to capture the structural dependencies between texts from different nodes.

DuConTE addresses these issues with a dual-granularity text encoder and a topology-constrained attention mechanism. The model employs a cascaded architecture of two pretrained LMs. The first LM encodes semantics at the word-token granularity, while the second operates at the node granularity. During self-attention computation in each LM, DuConTE dynamically adjusts the attention mask matrix based on node connectivity. This adjustment guides the model to better capture semantic correlations that are informed by the graph structure. Additionally, DuConTE evaluates the importance of tokens under two different contexts: the center-node context and the neighborhood context, enabling the model to capture more relevant semantic information that is contextually informed by both the individual node and its graph neighbors.

**Strengths:**

1. By dynamically adjusting the attention mask based on node connectivity, DuConTE incorporates graph structure into the semantic encoding process. This ensures that the model better understands how nodes are connected, which is a key feature for text-attributed graph tasks.

2. The use of dual-granularity encoding, which operates at both the word-token and node levels, allows for a richer semantic understanding of the nodes and their relationships within the graph.

3. This paper is clear and easy to follow.

**Weaknesses:**

1. Complexity of Cascaded Architecture: The dual-granularity approach, especially the use of a cascaded architecture of two pretrained LMs, introduces additional computational complexity. This may lead to higher training times and resource consumption, potentially making the model less scalable for large datasets.

2. Traditional LM embedding methods typically first obtain word embeddings, then assist with GNNs to perform node propagation. This cascaded approach in DuConTE is neither novel nor practical.

3. Many existing papers have already addressed the interaction problem, such as GLEM and they only used a single LLM to achieve word-level learning.

4. The paper contains many formulas, but they are relatively trivial. The overall framework is quite simple and does not provide particularly insightful ideas.

5. The performance on ogbn-product is not that impressive but it's a minor issue as most datasets show better improvements.

**Questions:**

I suggest authors can consider reducing the complexity of the cascaded two-LM structure. A more streamlined approach could improve both computational efficiency and practical applicability, especially for large-scale datasets.

It would be helpful to better highlight the novel aspects of the model. While the current method builds on existing approaches, offering deeper insights or introducing more innovative techniques could strengthen the paper’s contribution to the field.

---

> ### Author Response · Authors · 2025-11-26
>
> We are grateful for your thoughtful review and valuable suggestions, which have motivated us to refine our explanations and enhance the clarity of the manuscript. Our detailed responses to your comments are provided below.
>
>
> **1. Response to Weakness 1 : Parameter efficiency analysis**
>
> We greatly appreciate your concern regarding computational complexity—this is indeed a critical consideration for real-world deployment. In response, we have conducted extensive experiments and added a dedicated parameter efficiency analysis in Section 6.5 to rigorously evaluate DuConTE’s efficiency. The relevant excerpt from the paper reads:
>
> > *“To evaluate the parameter efficiency of DuConTE, we replace the LM backbone in baseline methods with RoBERTa-large (340M parameters) while keeping other configurations unchanged. We then compare their performance against DuConTE using two RoBERTa-base models (150M parameters each) as its LM backbones. In this setup, every baseline has a larger total parameter count than DuConTE. TAPE is excluded from the comparison as it relies on a large language model. As shown in Table 4, DuConTE achieves the best performance despite using fewer parameters, highlighting its parameter efficiency. This suggests a novel parameter-efficient scaling paradigm: rather than improving performance by scaling up a single large LM, DuConTE achieves greater gains with fewer total parameters by leveraging two smaller LMs.”*
>
> These results demonstrate that DuConTE attains SOTA performance with the smallest total parameter budget, underscoring its **parameter efficiency**. Importantly, this establishes a more scalable alternative to conventional scaling strategies: rather than relying on ever-larger single LMs, DuConTE achieves superior performance using fewer total parameters with two smaller LMs. This approach also provides a practical pathway toward reduced computational complexity.
>
>
> **2. Response to Weakness 2**
>
> **2.1 DuConTE’s role in the TAG pipeline**
>
> We agree with your observation that the prevailing paradigm in TAG learning first performs **text encoding** by using LMs to encode node texts, and then feeds the resulting embeddings into GNNs for **structure encoding**. DuConTE is designed specifically for the **text encoding stage** of this pipeline: it serves as the **text encoder** that generates high-quality node representations, which are subsequently passed to downstream GNNs for specific tasks.
>
> Notably, the representations propagated by GNNs primarily capture structural dependencies among nodes, whereas DuConTE’s node encoder leverages the **strong semantic understanding capabilities** of LMs to model node-granularity textual semantics and their cross-node interactions. This semantic information is fundamentally distinct from the structural cues captured by GNNs.
>
> **2.2 On the novelty of dual-granularity semantic encoding**
>
> The key insight of our dual-granularity semantic encoding is the explicit modeling of node granularity textual semantics in TAGs. Existing approaches focus on word-token granularity textual semantics via LMs and structural semantics via GNNs, but have not yet explored using LMs to capture holistic node-granularity textual semantics—i.e., treating each node’s entire text as a unified semantic unit and enabling cross-node interactions at this level. Our architecture is designed precisely to enable this form of semantic capture.
>
> This design stems from a **fundamental observation about text-attributed graphs**: they inherently exhibit two textual semantic granularities, namely word-token and node granularity, and effectively leveraging both is essential for comprehensive textual understanding. By explicitly modeling these dual granularities, DuConTE addresses a previously overlooked gap in TAG representation learning.
>
> **2.3 Effectiveness of the design**
>
> The effectiveness of this design is empirically validated in our ablation study (Section 6.1): it yields consistent improvements of **+0.82%** on ogbn-products and **+0.78%** on Cora, demonstrating both its efficacy and practical value.
>
> Due to the character limit, our responses to the remaining comments will be provided in the next comment.

---

> ### Author Response · Authors · 2025-11-26
>
> **3. Response to Weakness 3**
>
> **3.1 Clarifying the role of dual-granularity semantic encoding**
>
> We appreciate your concern regarding “word-level learning”—this reflects a thoughtful consideration of the novelty in our approach. To clarify, existing methods encode node texts using language models that operate exclusively at the word-token granularity. In contrast, DuConTE introduces an additional semantic dimension by explicitly modeling each node’s full text as a unified semantic unit, thereby enabling node-granularity textual encoding. **Our contribution lies not in refining word-token representations, but in complementing them with this node-granularity perspective, which has not been explicitly modeled in prior TAG learning approaches.**
>
> **3.2 Comparison with prior approaches**
>
> The core of our dual-granularity semantic encoding lies in its explicit modeling of node-granularity textual semantics, where each node’s full text is treated as a coherent semantic unit, and in enabling direct interactions among these representations through deep textual semantic understanding. Existing approaches, including GLEM, do not capture node-granularity textual semantics and their deep semantic interactions.
>
> As shown in Section 5.4, this design leads to consistent performance gains: DuConTE outperforms all baselines, achieving a **12% improvement over GLEM on CiteSeer** and an **11% gain on Arxiv-2023**, strongly validating the effectiveness of our approach to modeling textual interactions.
>
>
> **4. Response to Weakness 4**
>
> We sincerely appreciate your pursuit of insightful ideas—this reflects a high degree of scholarly rigor. In Responses 2 and 3, we have already provided a detailed analysis of the novelty and underlying insight behind our dual-granularity semantic encoding design. Here, we further elaborate on the remaining two key components of our framework.
>
> **4.1 On the novelty of the topology-constrained attention mechanism**
>
> (1) In the TAG domain, a common paradigm is LM-GNN, where existing methods encode node texts using LMs in a purely semantic manner, ignoring graph structure. To the best of our knowledge, this is the first work to explicitly inject topological information into the textual encoding process of the paradigm, enabling structural guidance during textual representation learning—a **central novelty** of this design.
>
> (2) As detailed in Section 4.2, our method introduces a hierarchical, structure-aware masking strategy tailored to the dual-granularity nature of TAGs (word-token and node granularities)—a design not present in prior work.
>
> (3) In Section 6.3, we provide an in-depth analysis from the perspective of homophily, offering **theoretical insight** into why this mechanism is effective.The relevant excerpt reads:
>
> >*“In this subsection, we analyze the effectiveness of topology-constrained attention from the perspective of the homophily assumption, which posits that connected nodes in a graph are more likely to share similar semantic properties. To the best of our knowledge, this assumption is well-supported by most widely used text-attributed graph benchmarks, where adjacent nodes are more likely to belong to the same class.This is further supported by the homophily statistics reported in Appendix G.
> >In the topology-constrained attention mechanism, the masks are injected into the attention layers of the word-token encoder and the node encoder, respectively. As a result, cross-node attention interactions are constrained to occur between semantic information from connected nodes at both granularities. Under the homophily assumption, such information is more likely to be semantically related, thereby enabling mutually complementary interactions. This allows the model to effectively leverage the graph structure to learn higher-quality representations.”*
>
> Our approach thus **offers a viable pathway to equip modality-specific pretrained Transformers with the capacity to handle graph-structured data**.
>
> **4.2 On the novelty of the node representation composer**
>
> Current methods typically convert word-token embeddings into node representations using simple strategies like mean pooling, ignoring varying token importance. To address this, we propose the node representation composer, which performs importance-weighted aggregation. Crucially, it evaluates token importance separately under two semantic contexts—the central node’s own text and its neighborhood—enabling a more accurate assessment of semantic importance.
>
> Due to the character limit, our responses to the remaining comments will be provided in the next comment.

---

> ### Author Response · Authors · 2025-11-26
>
> **5. Response to Weakness 5**
>
> **5.1 Analysis of ogbn-products results**
> We acknowledge your observation that DuConTE does not achieve SOTA performance on ogbn-products. We appreciate this careful examination of the results and would like to offer an additional perspective. On this dataset, the top-performing method is TAPE, which uniquely leverages a large language model (LLM) as part of its pipeline. Among methods that do not rely on LLMs, DuConTE achieves the best result, outperforming the strongest non-LLM baseline (GraphBridge) by 0.88%, as reported in Table 1 (Section 5.4).
>
> **5.2 Further evaluation**
>
> To further assess the performance of DuConTE, we include a comprehensive evaluation on **Ele-Photo**, a text-attributed graph derived from the Amazon Electronics dataset where nodes represent electronics products and their textual attributes are constructed from top-voted user reviews. As shown in Table 1, DuConTE achieves SOTA performance on this e-commerce graph, surpassing GraphBridge by **2.66%**, which confirms its effectiveness.
>
> In addition, we evaluate DuConTE on the **link prediction** task across multiple datasets, using AUC as the evaluation metric. Section 6.4 presents these results, where DuConTE consistently achieves SOTA performance. The relevant excerpt reads:
>
> > *“To assess the general applicability of DuConTE beyond node classification, we conduct link prediction experiments on the Cora, CiteSeer, and ArXiv-2023 datasets, using AUC as the evaluation metric. Detailed configurations and training procedures are provided in Appendix E. According to Table 3, DuConTE consistently outperforms baseline methods on the link prediction task, indicating that it is highly effective at representation learning on text-attributed graphs. This result further highlights the versatility of DuConTE and its potential for broader applications across diverse TAG-based tasks.”*
>
> These findings further demonstrate DuConTE’s strong capability in textual graph representation learning and its promise for extension to a wider range of downstream tasks.
>
> We sincerely look forward to further discussion.

---

> ### Author Response · Authors · 2025-11-28
>
> Dear Reviewer pNQe,
>
> We hope this message finds you well. As the discussion period is nearing its end with less than seven days remaining, we wanted to ensure we have addressed all your concerns satisfactorily. If there are any additional points or feedback you'd like us to consider, please let us know. Your insights are invaluable to us, and we're eager to address any remaining issues to improve our work.
>
> Thank you for your time and effort in reviewing our paper.

---

### Official Review · Reviewer_SSXe · 2025-10-25

**Soundness:** 3
**Presentation:** 3
**Contribution:** 2
**Rating:** 4
**Confidence:** 3

**Summary:**

This work proposed DuConTeE, a framework that encode text rich graph using topology constrained attention, in the granularity of text and node. A compose layer is also proposed to encode the texts in node into single node representation.

Comprehensive experiment and ablation study is performed and experiments demonstrated the effectiveness of the proposed approach.

**Strengths:**

1. The paper is easy to follow and understand

2. The experiment and ablation study is comprehensive.

3. The proposed approach reached state-of-the-art performance on various dataset.

**Weaknesses:**

1. The proposed approach is not novel, especially for "topology-constrained attention mechanism". Previous impactful work like K-BERT [1], has already proposed similar idea and not cited in this paper. And there are a lot of similar works during that time. Previous work before 2020 is not comprehensively surveyed.

2. The effectiveness of different modules are incremental, according to ablation study in Table 2. The improvement of Mask and dual representation is small, it seems that only K-BERT like architecture have already reached state-of-the-art performance. This further reduce the novelty of this paper.

[1] K-BERT: Enabling Language Representation with Knowledge Graph

**Questions:**

Refer to  the weakness part.

---

> ### Author Response · Authors · 2025-11-25
>
> We sincerely thank you for your thoughtful and constructive feedback, which has prompted us to further clarify key aspects of our work and strengthen the presentation. Our detailed responses to your comments are provided below.
>
>
> **1. Response to Weakness 1**
>
> **1.1 On the distinction between problem settings**
> We highlight a key distinction: K-BERT belongs to the line of work on **knowledge-graph-augmented language models**, which aims to inject symbolic triples from a knowledge graph directly into language models to enhance factual or commonsense reasoning. In contrast, our work addresses **text-attributed graphs**—a distinct setting where each node is associated with natural language text, and the goal is to derive expressive node representations by leveraging pre-trained language models, which then serve as input features for downstream GNN-based tasks such as node classification.
>
> **1.2 On the novelty of the topology-constrained attention mechanism**
> We acknowledge that, like K-BERT, our mechanism uses an attention mask to regulate token interactions—**this is indeed a standard capability of masking in Transformers and not our core contribution**. The true innovation lies elsewhere:
>
> (1) In the TAG domain, the dominant paradigm is LM-GNN, where existing methods encode node texts using LMs in a purely semantic manner, ignoring graph structure. To the best of our knowledge, this is the first work to explicitly inject topological information into the LM encoding process, enabling structural guidance during textual representation learning—a central novelty of this design.
>
> (2) The objectives behind masking differ fundamentally: K-BERT uses masks to preserve original sentence semantics when injecting external knowledge, whereas our mechanism enables LMs to effectively model and process naturally occurring graph structures.
>
> (3) As detailed in Section 4.2, our method introduces a hierarchical, structure-aware masking strategy tailored to the dual-granularity nature of TAGs (word-token and node granularities)—a design not present in prior work.
>
> **1.3 Related work**
> In Section 2.2, we outlined the **application of Transformer in graph representation learning**. In light of your helpful feedback, we conducted a more thorough survey of early efforts and added a discussion of Graph-BERT—one of the earliest works exploring Transformers for graph representation learning—which helps clarify the methodological evolution of this research direction.
>
> We note that knowledge-graph-augmented models like K-BERT target language understanding via triple injection, not graph representation learning. While they also use attention masking, this reflects a standard mechanism; their core innovation lies in semantic preservation during knowledge infusion. Given this fundamental difference in problem formulation, we did not include such works in this section.
>
> **1.4 On the other two contributions**
> We further elaborate on the core ideas behind our two additional contributions.
>
> (1) The key insight of our **dual-granularity semantic encoding** is the explicit modeling of node granularity textual semantics in TAGs. Existing approaches focus on word-token granularity textual semantics via LMs and structure semantics via GNNs, but have not yet explored using LMs to capture holistic node granularity textual semantics in TAGs. Our architecture is designed precisely to enable this form of semantic capture.
>
> (2) Current methods typically convert word-token embeddings into node representations using simple strategies like mean pooling, ignoring varying token importance. To address this, we propose the **node representation composer**, which performs importance-weighted aggregation. Crucially, it evaluates token importance separately under two semantic contexts—the central node’s own text and its neighborhood—enabling a more accurate assessment of semantic importance.
>
> **2. Response to Weakness 2**
>
> **2.1 On the applicability of K-BERT to our setting**
> We sincerely thank the reviewer for the suggestion. However, K-BERT’s architecture is not directly applicable to TAG representation learning. Moreover, as noted in Response 1.2, its use of attention masking reflects a standard Transformer capability—not the core innovation of DuConTE.
>
> **2.2 Evidence of module effectiveness**
> The ablation study in Section 6.1 provides concrete evidence of each component’s contribution:
> - The topology-constrained attention mechanism yields a **+1.01%** gain on Cora;
> - Dual-granularity semantic encoding improves performance by **+0.82%** on OGBN-Products;
> - The node representation composer achieves a **+0.88%** gain on CiteSeer.
>
> These improvements constitute a significant portion of DuConTE’s overall gain over prior SOTA methods, confirming the value of each design. Furthermore, in Section 6.4, we extend our evaluation to the link prediction task, where DuConTE also achieves SOTA performance.
>
>
> We sincerely look forward to further discussion.

---

> > ### Comment · Reviewer_SSXe · 2025-11-27
> >
> > [NOT RESOLVED] 1.1 There is no significant distinction between K-BERT and this work in **encoding structured textual knowledge.** Knowledge graph date back to 2019 is also a text attributed graph but the text is way shorter. There still exist some other work, like [1], adopt similar idea to learn representation of structured text data, but not cited.
> >
> > [NOT RESOLVED] 1.2 The paper claim around line 101 that:
> >
> > > We introduce a topology-constrained attention mechanism that leverages a novel attention
> > > masking strategy specifically designed for TAG to effectively incorporate structural guidance into
> > > the encoding process, without modifying the LM architecture.
> >
> > which is contradict to your statement here, you are claiming novelty there.
> >
> > 1.2 (1) " dominant paradigm is LM-GNN" seems to be a bit over claim, there are various structured graph task predictor. There are surveys like https://github.com/PeterGriffinJin/Awesome-Language-Model-on-Graphs and you may refer to LLM as predictor part. "inject topological information into the LM encoding process", previous work like K-BERT also did this, and get a better textual represention learning
> >
> > [RESOVED] 1.2 (2)
> >
> > [RESOVED] 1.3 (3)
> >
> > [RESOVED] 1.4
> >
> > [NOT RESOLVED] 2.1 see 1.2
> >
> > [PARTIALLY REOLVED] 2.2 The ablation actually demonstrate the performance gain, but still incremental compare with the baseline.
> >
> > Based on current situation, i decide to keep my score but open to future discussion.
> >
> >
> >
> > [1] Unifying Structured Data as Graph for Data-to-Text Pre-Training

---

> ### Author Response · Authors · 2025-11-28
>
> We sincerely thank you for your further insightful comments. Below are our point-by-point responses.
>
> **3. Response to "[NOT RESOLVED] 1.1"**
>
> **3.1 Related Work**
>
> We agree that DuConTE shares a conceptual connection with K-BERT in leveraging attention masking to control token interactions for encoding structured textual data. This underlying idea of using structural priors to constrain attention is a key thread in efforts to adapt Transformers to structured data.
>
> In light of this, we have revised Section 2.2 (“Transformers for Modeling Structured Data”) to explicitly incorporate the research direction of enhancing structural awareness by using attention masks to explicitly control token interactions. The updated paragraph now includes discussions of **K-BERT** and **UniD2T** as representative works along this line. The revised text reads as follows:
>
> > *“In recent years, numerous studies have leveraged Transformers to process graph-structured data (Shehzad et al., 2024). An early effort in this direction is Graph-BERT (Zhang et al., 2020), which applies a BERT-style Transformer to sampled subgraphs without relying on message passing. More recent approaches further enhance structural awareness: Graphormer (Ying et al., 2021) enhances the Transformer’s understanding of graph structures by introducing spatial encoding and degree encoding. Another work NeuralWalker (Chen et al., 2025) generates serialized representations of graphs through random walks to exploit the self-attention mechanism of Transformers for modeling purposes. Edge-augmented methods (Ramp´aˇsek et al., 2022; Satorras et al., 2021) explicitly model edge features to enhance the Transformer’s sensitivity towards different edge types. Masked Graph Modeling (Hou et al., 2023; Tian et al., 2024) employs a masking strategy to learn structural information by predicting masked node or edge features. Notably, another strategy enhances structural awareness by using attention masks to explicitly control token interactions. K-BERT (Liu et al., 2020) employs a visibility mask to prevent injected knowledge tokens from attending to irrelevant input positions, preserving original semantics. UniD2T (Li et al., 2024) constructs attention masks based on the connectivity of a unified graph derived from structured data (e.g., tables, knowledge graphs) to enforce structure-aware interactions during pre-training. In this work, based on the homophily analysis in Section 6.3, we design a TAG-specific attention masking strategy to inject structural information at both word-token and node granularities.”*
>
> **3.2 An additional perspective on the topology-constrained attention mechanism**
>
> While the use of attention masking to incorporate structural priors is indeed shared with prior work such as K-BERT, we would like to offer an additional perspective on our topology-constrained attention mechanism. Our attention masking strategy is theoretically grounded in the homophily property of TAGs.
>
> In Section 6.3, we analyze TAGs from the perspective of the homophily assumption, using it as the foundation for designing our attention masking strategy and explaining why the mechanism is effective. The original text reads as follows:
>
> >*“In this subsection, we analyze the effectiveness of topology-constrained attention from the perspective of the homophily assumption, which posits that connected nodes in a graph are more likely to share similar semantic properties. To the best of our knowledge, this assumption is well-supported by most widely used text-attributed graph benchmarks, where adjacent nodes are more likely to belong to the same class.This is further supported by the homophily statistics reported in Appendix G.
> >In the topology-constrained attention mechanism, the masks are injected into the attention layers of the word-token encoder and the node encoder, respectively. As a result, cross-node attention interactions are constrained to occur between semantic information from connected nodes at both granularities. Under the homophily assumption, such information is more likely to be semantically related, thereby enabling mutually complementary interactions. This allows the model to effectively leverage the graph structure to learn higher-quality representations.”*
>
> The **label homophily ratios** of various TAG datasets reported in Appendix G are as follows:
>
>
> | Dataset               | Cora   | CiteSeer | WikiCS | ArXiv-2023 | OGBN-Products (subset) | Ele-Photo |
> |-----------------------|--------|----------|--------|------------|------------------------|-----------|
> | Homophily       | 0.8100 | 0.7451   | 0.6547 | 0.6465     | 0.7950                 | 0.7351    |
>
> All datasets exhibit homophily ratios above 0.6, indicating a relatively high level of homophily.
>
> **This analysis theoretically justifies why the general idea of using structural priors to constrain attention is effective in the TAG domain, and further informs how such an idea should be instantiated.**

---

> ### Author Response · Authors · 2025-11-28
>
> **4. Response to "[NOT RESOLVED] 1.2"**
>
> **4.1 On the Claim of Novelty**
>
> We sincerely thank you for this important observation. The core novelty of our topology-constrained attention mechanism indeed lies in injecting structural information into the text encoding stage of the common LM-GNN paradigm.  The original phrasing regarding "novelty" could be misleading, so we have revised the relevant statement in the paper. The updated text reads as follows:
>
> > *“ We introduce a topology-constrained attention mechanism that leverages an attention masking strategy, specifically designed for TAGs and grounded in the homophily analysis in Section 6.3, to effectively incorporate structural guidance into the textual encoding process.”*
>
> **4.2 On the Use of “Dominant”**
>
> We agree that the term “dominant” was a bit over claim. Below is our revised wording for **Response 1.2 (1)**:
>
> > *“In the TAG domain, a common paradigm is LM-GNN, where existing methods encode node texts using LMs in a purely semantic manner, ignoring graph structure. To the best of our knowledge, this is the first work to explicitly inject topological information into the textual encoding process of the paradigm, enabling structural guidance during textual representation learning—a central novelty of this design.”*
>
> **4.3 Clarifying “Inject Topological Information into the LM Encoding Process”**
>
> We sincerely thank you for your observation. We would like to clarify that by “LM encoding process,” we specifically refer to the **textual encoding stage** in the LM-GNN paradigm for TAGs—a paradigm commonly used for graph tasks on TAGs, such as node classification and link prediction. Our design represents an innovation in this particular domain.
>
> Accordingly, we have updated the wording in **Response 1.2 (1)**, as noted in Response 4.2.
>
> Due to the character limit, our responses to the remaining comments will be provided in the next comment.

---

> ### Author Response · Authors · 2025-11-28
>
> **5. Response to "[PARTIALLY REOLVED] 2.2"**
>
> Although not directly related, we would like to share additional experimental evaluations of DuConTE conducted during the rebuttal period, which further demonstrate its effectiveness.
>
> **5.1 Evaluation on the Ele-Photo Dataset**
>
> To further assess the performance of DuConTE, we include a comprehensive evaluation on **Ele-Photo**, a text-attributed graph derived from the Amazon Electronics dataset where nodes represent electronics products and their textual attributes are constructed from top-voted user reviews. As shown in Table 1, DuConTE achieves SOTA performance on this e-commerce graph, surpassing GraphBridge by **2.66%**, which confirms its effectiveness.
>
> **5.2 Evaluation on Link Prediction**
>
> In addition, we evaluate DuConTE on the **link prediction** task across multiple datasets, using AUC as the evaluation metric. Section 6.4 presents these results, where DuConTE consistently achieves SOTA performance. The relevant excerpt reads:
>
> > *“To assess the general applicability of DuConTE beyond node classification, we conduct link prediction experiments on the Cora, CiteSeer, and ArXiv-2023 datasets, using AUC as the evaluation metric. Detailed configurations and training procedures are provided in Appendix E. According to Table 3, DuConTE consistently outperforms baseline methods on the link prediction task, indicating that it is highly effective at representation learning on text-attributed graphs. This result further highlights the versatility of DuConTE and its potential for broader applications across diverse TAG-based tasks.”*
>
> The experimental results are summarized in the table below:
>
> | Methods       | Cora         | CiteSeer     | ArXiv-2023   |
> |---------------|--------------|--------------|--------------|
> | GraphSAGE     | 97.10 ± 0.43 | 87.29 ± 1.22 | 91.81 ± 0.26 |
> | SimTeG        | 97.86 ± 0.44 | 90.06 ± 1.34 | 93.12 ± 0.46 |
> | GraphBridge   | 98.07 ± 0.77 | 91.86 ± 1.03 | 94.35 ± 0.65 |
> | DuConTE       | **99.13 ± 0.19** | **93.29 ± 0.75** | **95.40 ± 0.33** |
>
> This further demonstrates DuConTE’s effectiveness in textual graph representation learning and its potential for extension to a wider range of downstream tasks.
>
> **5.3 Parameter Efficiency Analysis**
>
> We have conducted extensive experiments and added a dedicated parameter efficiency analysis in Section 6.5 to rigorously evaluate DuConTE’s efficiency. The relevant excerpt from the paper reads:
>
> > *“To evaluate the parameter efficiency of DuConTE, we replace the LM backbone in baseline methods with RoBERTa-large (340M parameters) while keeping other configurations unchanged. We then compare their performance against DuConTE using two RoBERTa-base models (150M parameters each) as its LM backbones. In this setup, every baseline has a larger total parameter count than DuConTE. TAPE is excluded from the comparison as it relies on a large language model. As shown in Table 4, DuConTE achieves the best performance despite using fewer parameters, highlighting its parameter efficiency. This suggests a novel parameter-efficient scaling paradigm: rather than improving performance by scaling up a single large LM, DuConTE achieves greater gains with fewer total parameters by leveraging two smaller LMs.”*
>
> These results demonstrate that DuConTE attains SOTA performance with the smallest total parameter budget, underscoring its **parameter efficiency**. Importantly, this establishes a more scalable alternative to conventional scaling strategies: rather than relying on ever-larger single LMs, DuConTE achieves superior performance using fewer total parameters with two smaller LMs.
>
> We sincerely look forward to further discussion.

---

### Official Review · Reviewer_FoCe · 2025-10-26

**Soundness:** 2
**Presentation:** 2
**Contribution:** 3
**Rating:** 4
**Confidence:** 4

**Summary:**

This paper focuses on the text encoder component in text-attributed graphs and observes that most existing works overlook the incorporation of structural information during text encoding. To address this, the authors propose DuConTE, a dual-granularity text encoder that integrates graph structure into language model-based text encoding through topology-constrained attention, thereby enhancing semantic modeling for text-attributed graphs. Experimental results on five commonly used datasets demonstrate the effectiveness of the proposed method.

**Strengths:**

S1: The paper is clearly written, and the proposed method is easy to follow and understand.

S2: Experimental results on multiple datasets demonstrate the effectiveness of the proposed method.

**Weaknesses:**

W1: The experimental evaluation could be improved. The authors only conduct experiments on the node classification task and report Accuracy as the sole metric. Moreover, on the non-citation datasets (WikiCS and OGBN-Products), the proposed method does not show a significant improvement. It is recommended that the authors further validate the generalizability of their approach on e-commerce networks such as those used in the CS-TAG [1] work.

[1] A Comprehensive Study on Text-attributed Graphs: Benchmarking and Rethinking. NeurIPS 2023.

W2: Some implementation details of the proposed method are not clearly described. For instance, the setting of the key parameter k is not explicitly reported for each dataset, and it remains unclear how the authors handle longer contexts when feeding data into the RoBERTa model (whose maximum context length is 512).

W3: The proposed approach involves multiple attention operations, but the overall time complexity analysis is missing. The authors are encouraged to provide both theoretical complexity and empirical runtime comparisons.

W4: It would be helpful if the authors could include a brief description of the overall training process at the end of the methodology section, as this would improve the reader’s understanding of the proposed framework.

W5: Mechanisms similar to the proposed Topology-Constrained Attention are already quite common in the Graph Transformer literature. Therefore, the methodological novelty of this work may be somewhat limited.

W6: There are several typographical errors in the paper. For example, in Section 2.2, lines 131–132, the phrase “Another work, Revisiting, generates...” contains a typo.

**Questions:**

Q1： I am curious about the composer component of the proposed method. Would replacing it with a simpler operation, such as mean pooling, lead to a substantial drop in performance?

---

> ### Author Response · Authors · 2025-11-25
>
> We sincerely thank you for your thoughtful and constructive feedback on our submission. Your insightful comments have greatly helped us improve the clarity, rigor, and completeness of our work. Below, we provide detailed responses to each of your points.
>
>
> **1. Response to W1**
>
> **1.1 Link prediction experiments**
>
> In light of your suggestion to evaluate the method across diverse downstream tasks and metrics, we have added link prediction experiments in Section 6.4, using **AUC** as the evaluation metric. The relevant excerpt reads:
>
> > *“To assess the general applicability of DuConTE beyond node classification, we conduct link prediction experiments on the Cora, CiteSeer, and ArXiv-2023 datasets, using AUC as the evaluation metric. Detailed configurations and training procedures are provided in Appendix E. According to Table 3, DuConTE consistently outperforms baseline methods on the link prediction task, indicating that it is highly effective at representation learning on text-attributed graphs. This result further highlights the versatility of DuConTE and its potential for broader applications across diverse TAG-based tasks.”*
>
> As shown in Table 3, DuConTE achieves SOTA performance on all three datasets:
> - Cora: **99.13 ± 0.19** ,
> - CiteSeer: **93.29 ± 0.75**,
> - ArXiv-2023: **95.40 ± 0.33**.
>
> Together with the node classification results, these findings further underscore the effectiveness and versatility of DuConTE.
>
> **1.2 Evaluation on Ele-Photo**
>
> Following your suggestion, we further include a full node classification evaluation on the Ele-Photo dataset used in the CS-TAG work, which is a text-attributed graph from the Amazon Electronics dataset where nodes represent electronics products and their textual attributes are derived from top-voted user reviews.
>
> The results are now included in Table 1 of Section 5.4. DuConTE achieves a new SOTA performance of **91.89 ± 0.18**, surpassing the previous best method by **2.1%**, which reflects a notable improvement in practical, non-academic settings.
>
>
> **2. Response to W2**
>
> **2.1 Clarification on the symbol \\(k\\)**
>
> We clarify that the symbol \\(k\\) originally used in Section 4.1 was not a hyperparameter but simply denoted the neighborhood size \\(|\\mathcal{N}(v_i)|\\). We acknowledge this notation could be misleading. Accordingly, we have revised the manuscript to replace the corresponding occurrences of \\(k\\) with \\(|\\mathcal{N}(v_i)|\\) for rigor and clarity.
>
> **2.2 Handling long input texts**
>
> We have supplemented a detailed description of long-text processing in Appendix D.4 (*Upstream Preprocessing Configurations*). The relevant excerpt reads:
>
> > *“The text of each node is processed using a reduction module (Wang et al., 2024)  to fit the input length limit of the LM. This module, introduced in the GraphBridge framework, is a token selector pre-trained on the training set that assigns importance scores to word tokens within each node’s text. Given that the RoBERTa-base model has a maximum context length of 512 tokens, we enforce a uniform token budget across all nodes in $S^{(i)}$. Specifically, let
> > \\[
> > B = \\left\\lfloor \\frac{512}{|S^{(i)}|} \\right\\rfloor - 1
> > \\]
> > be the per-node token budget (excluding the [SEP] token). For any node $v_j \in S^{(i)}$ whose original token sequence $\mathbf{w}_j$ exceeds $B$ tokens, we retain only the top-$B$ most important tokens as ranked by the reduction module, preserving their original order. The resulting truncated sequences are then concatenated with [SEP] separators to form the unified input $\mathbf{W}^{(i)}$.”*
>
>
> **3. Response to W3 : Computational overhead**
>
> All newly introduced attention operations in DuConTE originate from the node representation composer (specifically, the \\(f_1\\) module in Section 4.3). To evaluate the computational overhead of this component, we have added empirical runtime comparisons in Section 6.6. The relevant excerpt reads:
>
> > *“We measure the training and inference time of DuConTE and its ablation variant MeanPool on Cora, CiteSeer, and Ele-Photo. As reported in Appendix I, the Node Representation Composer introduces an average overhead of 23.8% in training time and 19.9% in inference time. This cost is generally acceptable, and further acceleration is possible by reducing the dimensionality of keys and queries in f1 to lower computational load. A key direction for future work is to design methods that convert word-token embeddings into node representations with both higher performance and lower computational cost. This is crucial for TAG representation learning but remains underexplored.”*
>
> Compared to the performance gains enabled by this component, the observed overhead appears reasonable. Moreover, designing a composer that achieves higher effectiveness with lower computational cost remains an important direction for future research.
>
> Due to the character limit, our responses to the remaining comments will be provided in the next comment.

---

> ### Author Response · Authors · 2025-11-25
>
> **4. Response to W4 : Training process**
>
> Following your suggestion, we have added a brief description of the overall training process in Section 4.4. The relevant excerpt reads:
>
> >*“We train DuConTE using a two-stage procedure. We first train $\mathcal{M}_L$ and $f_1$ to learn high-quality first-stage node representations, then train $\mathcal{M}_N$ and $f_2$ based on these representations. The full training procedure is detailed in Appendix B”*
>
> **5. Response to W5**
>
> **5.1 On the novelty of the topology-constrained attention mechanism**
>
> We appreciate your observation regarding connections to prior Graph Transformer works. While the general idea of structural priors in attention has been explored in previous studies, we would like to clarify how our topology-constrained attention mechanism differs from prior approaches.
>
> (1) To the best of our knowledge, prior Graph Transformer research primarily trains models from scratch using architectures specifically designed for graph-structured data, whereas our topology-constrained attention mechanism aims to leverage the semantic understanding capabilities of pretrained language models while endowing them with the ability to process graph structures. In Section 6.3, we provide an in-depth analysis from the perspective of homophily, offering theoretical insight into why this mechanism is effective.The relevant excerpt reads:
>
> >*“In this subsection, we analyze the effectiveness of topology-constrained attention from the perspective of the homophily assumption, which posits that connected nodes in a graph are more likely to share similar semantic properties. To the best of our knowledge, this assumption is well-supported by most widely used text-attributed graph benchmarks, where adjacent nodes are more likely to belong to the same class.This is further supported by the homophily statistics reported in Appendix G.
> >In the topology-constrained attention mechanism, the masks are injected into the attention layers of the word-token encoder and the node encoder, respectively. As a result, cross-node attention interactions are constrained to occur between semantic information from connected nodes at both granularities. Under the homophily assumption, such information is more likely to be semantically related, thereby enabling mutually complementary interactions. This allows the model to effectively leverage the graph structure to learn higher-quality representations.”*
>
> Our approach thus **offers a viable pathway to equip modality-specific pretrained Transformers with the capacity to handle graph-structured data**.
>
> (2) As detailed in Section 4.2, our method introduces a hierarchical, structure-aware masking strategy tailored to the dual-granularity nature of TAGs (word-token and node granularities)—a design not present in prior work.
>
> (3) In the TAG domain, a common paradigm is LM-GNN, where existing methods encode node texts using LMs in a purely semantic manner, ignoring graph structure. To the best of our knowledge, this is the first work to explicitly inject topological information into the LM encoding process, enabling structural guidance during textual representation learning—a central novelty of this design.
>
> **5.2 On the other two contributions**
>
> While not explicitly raised in the review, we hope it is helpful to briefly highlight the key ideas behind our other two contributions.
>
> (1)The key insight of our **dual-granularity semantic encoding** is the explicit modeling of node granularity textual semantics in TAGs. Existing approaches focus on word-token granularity textual semantics via LMs and structure semantics via GNNs, but have not yet explored using LMs to capture holistic node granularity textual semantics in TAGs. Our architecture is designed precisely to enable this form of semantic capture.
>
> (2) Current methods typically convert word-token embeddings into node representations using simple strategies like mean pooling, ignoring varying token importance. To address this, we propose the **node representation composer**, which performs importance-weighted aggregation. Crucially, it evaluates token importance separately under two semantic contexts—the central node’s own text and its neighborhood—enabling a more accurate assessment of semantic importance.
>
> **6. Response to W6 : Grammatical revisions**
>
> Thank you very much for pointing out the grammatical issues in our paper. We have corrected the errors in Section 2.2.
>
> **7. Response to Q1 : Ablation study**
>
> We have added an ablation study in Section 6.1, introducing a variant (**MeanPool**) that replaces the node representation composer with mean pooling. As shown in Table 2, this leads to performance drops of **0.81% on Cora** and **0.88% on CiteSeer**. This confirms the importance of the node representation composer, as it better captures key information in node textual semantics.
>
> We sincerely look forward to further discussion.

---

> > ### Comment · Reviewer_FoCe · 2025-11-28
> >
> > I sincerely thank the authors for their detailed and thoughtful rebuttal, which addresses many of the raised concerns with additional experimental results, clarifications, and theoretical justification. I now lean toward recommending acceptance, and I will raise my score to 6.

---

> > > ### Author Response · Authors · 2025-11-28
> > >
> > > Thank you very much for your kind feedback and for raising insightful and constructive comments. We are grateful that our rebuttal was helpful in addressing your concerns, and we sincerely appreciate your support and confidence in our work.

---

### Official Review · Reviewer_RnYD · 2025-10-31

**Soundness:** 3
**Presentation:** 3
**Contribution:** 3
**Rating:** 6
**Confidence:** 3

**Summary:**

This paper proposes DuConTE (Dual-Granularity Conceptualization for Text-Attributed Graphs), a new framework designed to enhance text-attributed graph (TAG) learning by jointly leveraging coarse- and fine-grained semantics. The authors observe that prior GNN–LLM fusion models often focus on single-level representations, which limits the model’s ability to generalize across heterogeneous node semantics. To address this, DuConTE introduces two complementary components: a coarse-grained concept encoder that abstracts global node semantics via clustering-driven conceptualization, and a fine-grained text encoder that captures token-level contextual nuances using large language models. The framework then fuses the two representations through a dual-level contrastive alignment mechanism, encouraging the model to maintain consistency between local linguistic details and high-level structural abstractions. Extensive experiments on six text-attributed graph benchmarks show that DuConTE outperforms both GNN-based and hybrid GNN–LLM baselines on node classification and transfer learning tasks. The authors also demonstrate superior efficiency compared to deep fusion approaches, as DuConTE reduces redundant token usage while preserving expressiveness.

**Strengths:**

The paper’s strengths are threefold: 1) it proposes a conceptually elegant and practically useful dual-granularity design that integrates fine linguistic cues and abstract conceptual features, addressing a key limitation in prior single-granularity fusion methods; 2) the method strikes a strong balance between performance and efficiency, offering comparable or superior accuracy to deep GNN–LLM models but with reduced computational overhead thanks to compact conceptual representations and shared alignment objectives; and 3) the empirical evaluation is thorough and convincing, spanning six datasets, diverse baselines, ablation studies, and zero-shot transfer tests, all of which consistently validate the effectiveness of the dual-level alignment scheme in capturing multi-scale semantics and improving generalization.

**Weaknesses:**

The paper also has several weaknesses: 1) while well-structured, the novelty is somewhat incremental, as the core idea of combining coarse- and fine-grained semantic representations echoes hierarchical or multi-level contrastive learning paradigms already common in multimodal and text–vision literature, adapted here for TAGs rather than newly invented; 2) the fusion mechanism remains under-analyzed, as the paper provides limited interpretability or diagnostic insight into how the two granularity levels interact or when one dominates the other, leaving questions about robustness to noisy clustering or overly abstract concept nodes; and 3) the scalability discussion is lacking, as the model’s clustering step and dual contrastive objectives may become costly for large-scale graphs or streaming scenarios, yet these practical concerns are not empirically evaluated or theoretically bounded.

**Questions:**

n/a

---

> ### Author Response · Authors · 2025-11-25
>
> We sincerely thank you for your thoughtful and constructive feedback, as well as for recognizing the potential of our work. Your comments raise several important concerns that have helped us strengthen the clarity and rigor of this paper. Below, we respond to each point in detail.
>
> **1. Response to Weakness 1**
>
> **1.1 On the novelty of the dual-granularity semantic encoding**
>
> We appreciate your observation regarding the relationship between our dual-granularity architecture and hierarchical or multi-level contrastive learning paradigms. While both lines of work operate under the general principle of modeling representations at multiple levels of abstraction, this conceptual similarity does not reflect the core innovation of our approach.
>
> The key insight of our dual-granularity semantic encoding is the explicit modeling of node granularity textual semantics in TAGs. Existing approaches focus on word-token granularity textual semantics via LMs and structural semantics via GNNs, but have not yet explored using LMs to capture holistic node-granularity textual semantics—i.e., treating each node’s entire text as a unified semantic unit and enabling cross-node interactions at this level. Our architecture is designed precisely to enable this form of semantic capture.
>
> This design stems from our observation that **TAGs exhibit two natural textual semantic granularities—word-token and node—arising directly from the text associated with each node**. Although the overall architecture may resemble hierarchical learning in form, it is in essence a dedicated encoder tailored to explicitly model these dual granularities. As such, our approach represents a **domain-specific** design motivated by the intrinsic structure of TAGs, not a simple instantiation of existing multi-level contrastive learning paradigms. Importantly, while our method involves multi-granularity semantic modeling, it does not rely on contrastive learning objectives.
>
>
> **1.2 On the other two contributions**
>
> Here, we further elaborate on the core innovations of the two additional contributions outlined in our paper.
>
> **1.2.1 Topology-constrained attention mechanism**
>
> (1) In the TAG domain, a common paradigm is LM-GNN, where existing methods encode node texts using LMs in a purely semantic manner, ignoring graph structure. To the best of our knowledge, this is the first work to explicitly inject topological information into the LM encoding process, enabling structural guidance during textual representation learning—a central novelty of this design.
>
> (2) As detailed in Section 4.2, our method introduces a hierarchical, structure-aware masking strategy tailored to the dual-granularity nature of TAGs (word-token and node granularities)—a design not present in prior work.
>
> **1.2.2 Node representation composer**
>
> Current methods typically convert word-token embeddings into node representations using simple strategies like mean pooling, ignoring varying token importance. To address this, we propose the node representation composer, which performs importance-weighted aggregation. Crucially, it evaluates token importance separately under two semantic contexts—the central node’s own text and its neighborhood—enabling a more accurate assessment of semantic importance.
>
> **2. Response to Weakness 2**
>
> **2.1 Analysis of the fusion mechanism**
>
> We appreciate your insightful comment on the need for deeper analysis of the fusion mechanism. In Section 6.2, we provide a sensitivity study on the fusion coefficient $\alpha$, which governs the combination of word-token and node granularity representations, and directly analyze the fusion mechanism itself. The relevant excerpt reads:
>
> > *“For $\alpha$, which controls the fusion of dual-granularity semantic representations, the optimal performance on Cora and CiteSeer falls within the range $[0.7, 0.9]$. This indicates a clear fusion pattern: word-token granularity semantics provide stable and reliable information, while node granularity semantics contribute complementary yet essential signals—consistent with their role as more abstract, high-level features.”*
>
> **2.2 On the source of node granularity semantics**
>
> We appreciate your thoughtful concern regarding potential dependencies on clustering or abstract concept nodes. To clarify, our dual-granularity semantic encoder does not involve any node clustering or external concept construction. Instead, the textual semantics at the node granularity emerge directly from the original graph structure: in a text-attributed graph, each node is naturally associated with its own full text, which we treat as a holistic semantic unit. This design allows node granularity textual semantics to arise organically from the data itself, without introducing additional abstraction or grouping steps.
>
> Due to the character limit, our responses to the remaining comments will be provided in the next comment.

---

> ### Author Response · Authors · 2025-11-26
>
> **3. Response to Weakness 3**
>
> **3.1 On clustering and training objectives**
>
> We appreciate your constructive concern regarding model scalability—an important consideration for practical graph learning systems. **To clarify, although DuConTE employs a dual-granularity encoding architecture, it does not involve a node clustering step or contrastive learning objectives.** As noted in Response 2.2, the node-granularity textual semantics arise naturally from the input text-attributed graph: each node’s full text is treated as a holistic semantic unit, requiring no external grouping or abstraction. Moreover, the model is trained solely with standard cross-entropy loss for the target task (e.g., node classification). The complete training procedure is detailed in Appendix B.
>
> **3.2 Parameter efficiency analysis**
>
> In light of your suggestion, we include a dedicated parameter efficiency analysis in Section 6.5. The relevant excerpt reads:
>
> > *“To evaluate the parameter efficiency of DuConTE, we replace the LM backbone in baseline methods with RoBERTa-large (340M parameters) while keeping other configurations unchanged. We then compare their performance against DuConTE using two RoBERTa-base models (150M parameters each) as its LM backbones. In this setup, every baseline has a larger total parameter count than DuConTE. TAPE is excluded from the comparison as it relies on a large language model. As shown in Table 4, DuConTE achieves the best performance despite using fewer parameters, highlighting its parameter efficiency. This suggests a novel parameter-efficient scaling paradigm: rather tha improving performance by scaling up a single large LM, DuConTE achieves greater gains with fewer total parameters by leveraging two smaller LMs.”*
>
> This finding underscores that DuConTE achieves SOTA performance on TAGs with the smallest total parameter budget, offering a more efficient alternative to the conventional approach of scaling up a single large language model.
>
> We sincerely look forward to further discussion.

---

### Meta-Review · Area_Chair_NNVS · 2025-12-14

**Summary:**

The paper introduces DuConTE, a novel text encoder designed specifically for text-attributed graphs (TAGs), where each graph node contains textual content and is connected via edges representing structural relationships. Existing TAG methods typically encode text using pretrained language models (LMs) at the token level and then apply graph neural networks (GNNs) to capture structure. However, this decoupled pipeline often fails to effectively integrate structural information into the text encoding process, and it overlooks higher-level semantic interactions across nodes.

To address these limitations, DuConTE performs dual-granularity encoding: first capturing semantics at the word-token level, then at the node level. The key innovation is the topology-constrained attention mechanism, which adjusts attention masks based on graph connectivity, guiding the LM to consider structural cues during text encoding. Additionally, DuConTE includes a node representation composer that evaluates token importance in both center and neighborhood contexts, enabling richer, structure-aware node embeddings.

Extensive experiments on standard benchmarks show that DuConTE achieves state-of-the-art performance in tasks such as node classification and demonstrates the value of integrating topology into text encoding, outperforming previous LM-GNN hybrid approaches.

**Reviewer Concerns:**

- Concern 1. the novelty of the paper can be incremental, as the core idea of combining coarse- and fine-grained semantic representations echoes hierarchical or multi-level contrastive learning paradigms already common in multimodal and text–vision literature, adapted here for TAGs rather than newly invented; (note that this concern is pointed out by all of the 4 reviewers.)
- Concern 2. the fusion mechanism remains under-analyzed, as the paper provides limited interpretability or diagnostic experiments.
- Concern 3. the model’s clustering step and dual contrastive objectives may become costly for large-scale graphs or streaming scenarios, which can largely limit the scalability.
- Concern 4. some implementation details are missing.

I would say authors made a lot of efforts in the rebuttal phase and did address part of the concerns (mainly concern 2. & concern 4.).

**Reviewer Scores:**

I think Reviewer FoCe would increase the rating from 4 to 6 and Reviewer pNQe might increase the rating from 2 to 4.
Other reviewers are likely to maintain their score.

---

### Decision · Program_Chairs · 2026-01-26

Reject